# FACTORIZED FOURIER NEURAL OPERATORS

**Alasdair Tran**[1]        **Alexander Mathews** [1]        **Lexing Xie** [1]        **Cheng Soon Ong** [1,2]

[1] Australian National University        [2] Data61, CSIRO

## ABSTRACT

We propose the Factorized Fourier Neural Operator (F-FNO), a learning-based approach for simulating partial differential equations (PDEs). Starting from a recently proposed Fourier representation of flow fields, the F-FNO bridges the performance gap between pure machine learning approaches to that of the best numerical or hybrid solvers. This is achieved with new representations – separable spectral layers and improved residual connections – and a combination of training strategies such as the Markov assumption, Gaussian noise, and cosine learning rate decay. On several challenging benchmark PDEs on regular grids, structured meshes, and point clouds, the F-FNO can scale to deeper networks and outperform both the FNO and the geo-FNO, reducing the error by 83% on the Navier-Stokes problem, 31% on the elasticity problem, 57% on the airfoil flow problem, and 60% on the plastic forging problem. Compared to the state-of-the-art pseudo-spectral method, the F-FNO can take a step size that is an order of magnitude larger in time and achieve an order of magnitude speedup to produce the same solution quality.

## 1 INTRODUCTION

From modeling population dynamics to understanding the formation of stars, partial differential equations (PDEs) permeate the world of science and engineering. For most real-world problems, the lack of a closed-form solution requires using computationally expensive numerical solvers, sometimes consuming millions of core hours and terabytes of storage (Hosseini et al., 2016). Recently, machine learning methods have been proposed to replace part (Kochkov et al., 2021) or all (Li et al., 2021a) of a numerical solver.

Of particular interest are Fourier Neural Operators (FNOs) (Li et al., 2021a), which are neural networks that can be trained end-to-end to learn a mapping between infinite-dimensional function spaces. The FNO can take a step size much bigger than is allowed in numerical methods, can perform super-resolution, and can be trained on many PDEs with the same underlying architecture. A more recent variant, dubbed geo-FNO (Li et al., 2022), can handle irregular geometries such as structured meshes and point clouds. However, this first generation of neural operators suffers from stability issues. Lu et al. (2022) find that the performance of the FNO deteriorates significantly on complex geometries and noisy data. In our own experiments, we observe that both the FNO and the geo-FNO perform worse as we increase the network depth, eventually failing to converge at 24 layers. Even at 4 layers, the error between the FNO and a numerical solver remains large (14% error on the Kolmogorov flow).

In this paper, we propose the Factorized Fourier Neural Operator (F-FNO) which contains an improved representation layer for the operator, and a better set of training approaches. By learning features in the Fourier space in each dimension independently, a process called Fourier factorization, we are able to reduce the model complexity by an order of magnitude and learn higher-dimensional problems such as the 3D plastic forging problem. The F-FNO places residual connections after activation, enabling our neural operator to benefit from a deeply stacked network. Coupled with training techniques such as teacher forcing, enforcing the Markov constraints, adding Gaussian noise to inputs, and using a cosine learning rate scheduler, we are able to outperform the state of the art by a large margin on three different PDE systems and four different geometries. On the Navier-Stokes (Kolmogorov flow) simulations on the torus, the F-FNO reduces the error by 83% compared to the FNO, while still achieving an order of magnitude speedup over the state-of-the-art pseudo-spectral method (Figs. 3 and 4). On point clouds and structured meshes, the F-FNO outperforms the geo-FNO on both structural mechanics and fluid dynamics PDEs, reducing the error by up to 60% (Table 2).

Overall, we make the following three key contributions:

1. We propose a new representation, the F-FNO, which consists of separable Fourier representation and improved residual connections, reducing the model complexity and allowing it to scale to deeper networks (Fig. 2 and Eqs. (7) and (8)).

2. We show the importance of incorporating training techniques from the existing literature, such as Markov assumption, Gaussian noise, and cosine learning rate decay (Fig. 3); and investigate how well the operator can handle different input representations (Fig. 5).

3. We demonstrate F-FNO's strong performance in a variety of geometries and PDEs (Fig. 3 and Table 2). Code, datasets, and pre-trained models are available[1].

## 2 RELATED WORK

Classical methods to solve PDE systems include finite element methods, finite difference methods, finite volume methods, and pseudo-spectral methods such as Crank-Nicholson and Carpenter-Kennedy. In these methods, space is discretized, and a more accurate simulation requires a finer discretization which increases the computational cost. Traditionally, we would use simplified models for specific PDEs, such as Reynolds averaged Navier-Stokes (Alfonsi, 2009) and large eddy simulation (Lesieur & Métais, 1996), to reduce this cost. More recently, machine learning offers an alternative approach to accelerate the simulations. There are two main clusters of work: hybrid approaches and pure machine learning approaches. Hybrid approaches replace parts of traditional numerical solvers with learned alternatives but keep the components that impose physical constraints such as conservation laws; while pure machine learning approaches learn the time evolution of PDEs from data only.

**Hybrid methods**   typically aim to speed up traditional numerical solvers by using lower resolution grids (Bar-Sinai et al., 2019; Um et al., 2020; Kochkov et al., 2021), or by replacing computationally expensive parts of the solver with learned alternatives Tompson et al. (2017); Obiols-Sales et al. (2020). Bar-Sinai et al. (2019) develop a data driven method for discretizing PDE solutions, allowing coarser grids to be used without sacrificing detail. Kochkov et al. (2021) design a technique specifically for the Navier-Stokes equations that uses neural network-based interpolation to calculate velocities between grid points rather than using the more traditional polynomial interpolation. Their method leads to more accurate simulations while at the same time achieving an 86-fold speed improvement over Direct Numerical Simulation (DNS). Similarly, Tompson et al. (2017) employ a numerical solver and a decomposition specific to the Navier-Stokes equations, but introduce a convolutional neural network to infer the pressure map at each time step. While these hybrid methods are effective when designed for specific equations, they are not easily adaptable to other PDE tasks.

An alternative approach, less specialized than most hybrid methods but also less general than pure machine learning methods, is learned correction (Um et al., 2020; Kochkov et al., 2021) which involves learning a residual term to the output of a numerical step. That is, the time derivative is now $\mathbf{u}_t = \mathbf{u}_t^* + LC(\mathbf{u}_t^*)$, where $\mathbf{u}_t^*$ is the velocity field provided by a standard numerical solver on a coarse grid, and $LC(\mathbf{u}_t^*)$ is a neural network that plays the role of super-resolution of missing details.

**Pure machine learning approaches**   eschew the numerical solver altogether and learn the field directly, i.e., $\mathbf{u}_t = \mathcal{G}(\mathbf{u}_{t-1})$, where $\mathcal{G}$ is dubbed a neural operator. The operator can include graph neural networks Li et al. (2020a;b), low-rank decomposition Kovachki et al. (2021b), or Fourier transforms (Li et al., 2021a;b). Pure machine learning models can also incorporate physical constraints, for example, by carefully designing loss functions based on conservation laws (Wandel et al., 2020). They can even be based on existing simulation methods such as the operator designed by Wang et al. (2020) that uses learned filters in both Reynolds-averaged Navier-Stokes and Large Eddy Simulation before combining the predictions using U-Net. However, machine learning methods need not incorporate such constraints – for example, Kim et al. (2019) use a generative CNN model to represent velocity fields in a low-dimensional latent space and a feedforward neural network to advance the latent space to the next time point. Similarly, Bhattacharya et al. (2020) use PCA to map from an infinite dimensional input space into a latent space, on which a neural network operates before being transformed to the output space. Our work is most closely related to the Fourier

---

[1]`https://github.com/alasdairtran/fourierflow`

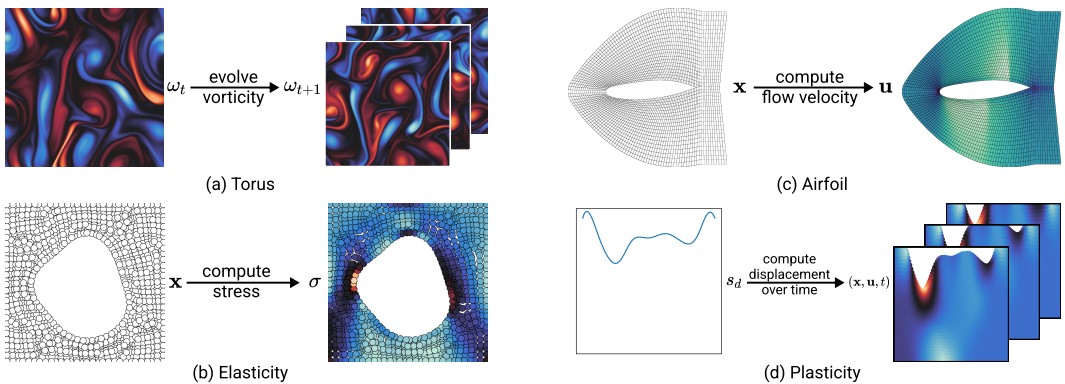

Figure 1: An illustration of the input and output of different PDE problems. See the accompanying Table 1 for details. On the torus datasets (a), the operator learns to evolve the vorticity over time. On Elasticity (b), the operator learns to predict the stress value on each point on a point cloud. On Airfoil (c), the operator learns to predict the flow velocity on each mesh point. On Plasticity (d), the operator learns the displacement of each mesh point given an initial boundary condition.

transform-based approaches (Li et al., 2021a; 2022) which can efficiently model PDEs with zero-shot super-resolution but is not specific to the Navier-Stokes equations.

**Fourier representations** are popular in deep-learning due to the efficiency of convolution operators in the frequency space, the $O(n \log n)$ time complexity of the fast Fourier transform (FFT), and the ability to capture long-range dependencies. Two notable examples of deep learning models that employ Fourier representations are FNet (Lee-Thorp et al., 2021) for encoding semantic relationships in text classification and FNO (Li et al., 2021a) for flow simulation. In learning mappings between function spaces, the FNO outperforms graph-based neural operators and other finite-dimensional operators such as U-Net. In modeling chaotic systems, the FNO has been shown to capture invariant properties of chaotic systems (Li et al., 2021b). More generally, Kovachki et al. (2021a) prove that the FNO can approximate any continuous operator.

## 3 THE FACTORIZED FOURIER NEURAL OPERATOR

**Solving PDEs with neural operators** An operator $\mathcal{G} : \mathcal{A} \to \mathcal{U}$ is a mapping between two infinite-dimensional function spaces $\mathcal{A}$ and $\mathcal{U}$. Exactly what these function spaces represent depends on the problem. In general, solving a PDE involves finding a solution $u \in \mathcal{U}$ given some input parameter $a \in \mathcal{A}$, and we would train a neural operator to learn the mapping $a \mapsto u$. Consider the vorticity formulation of the 2D Navier-Stokes equations,

$$\frac{\partial \omega}{\partial t} + \mathbf{u} \cdot \nabla \omega = \nu \nabla^2 \omega + f \qquad \nabla \cdot \mathbf{u} = 0 \tag{1}$$

where $\mathbf{u}$ is the velocity field, $\omega$ is the vorticity, and $f$ is the external forcing function. These are the governing equations for the torus datasets (Fig. 1a). The neural operator would learn to evolve this field from one time step to the next: $\omega_t \mapsto \omega_{t+1}$. Or consider the equation for a solid body in structural mechanics,

$$\rho \frac{\partial^2 \mathbf{u}}{\partial t^2} + \nabla \cdot \sigma = 0, \tag{2}$$

where $\rho$ is the mass density, $\mathbf{u}$ is the displacement vector and $\sigma$ is the stress tensor. Elasticity (Fig. 1b) and Plasticity (Fig. 1d) are both governed by this equation. In Plasticity, we would learn to map the initial boundary condition $s_d : [0, L] \to \mathbb{R}$ to the grid position $\mathbf{x}$ and displacement of each grid point over time: $s_d \mapsto (\mathbf{x}, \mathbf{u}, t)$. In Elasticity, we are instead interested in predicting the stress value for each point: $\mathbf{x} \mapsto \sigma$. Finally consider the Euler equations to model the airflow around an aircraft wing (Fig. 1c):

$$\frac{\partial \rho}{\partial t} + \nabla \cdot (\rho \mathbf{u}) = 0 \qquad \frac{\partial \rho \mathbf{u}}{\partial t} + \nabla \cdot (\rho \mathbf{u} \otimes \mathbf{u} + p\mathbb{I}) = 0 \qquad \frac{\partial E}{\partial t} + \nabla \cdot ((E + p)\mathbf{u}) = 0 \tag{3}$$

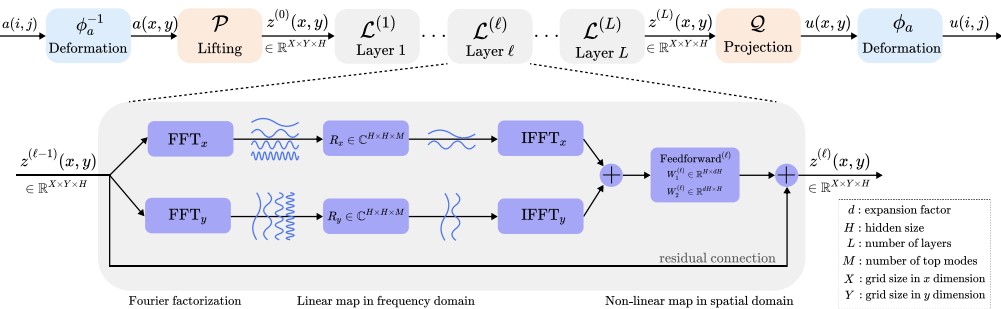

Figure 2: The architecture of the Factorized Fourier Neural Operator (F-FNO) for a 2D problem. The iterative process (Eq. (4)) is shown at the top, in which the input function $a(i, j)$ is first deformed from an irregular space into a uniform space $a(x, y)$, and is then fed through a series of operator layers $\mathcal{L}$ in order to produce the output function $u(i, j)$. A zoomed-in operator layer (Eq. (7)) is shown at the bottom which shows how we process each spatial dimension independently in the Fourier space, before merging them together again in the physical space.

where $\rho$ is the fluid mass density, $p$ is the pressure, $\mathbf{u}$ is the velocity vector, and $E$ is the energy. Here the operator would learn to map each grid point to the velocity field at equilibrium: $\mathbf{x} \mapsto \mathbf{u}$.

**Original FNO and geo-FNO architectures**  Motivated by the kernel formulation of the solution to linear PDEs using Green's functions, Li et al. (2020b; 2022) propose an iterative approach to map input function $a$ to output function $u$,

$$u = \mathcal{G}(a) = (\phi \circ \mathcal{Q} \circ \mathcal{L}^{(L)} \circ \cdots \circ \mathcal{L}^{(1)} \circ \mathcal{P} \circ \phi^{-1})(a), \tag{4}$$

where $\circ$ indicates function composition, $L$ is the number of layers/iterations, $\mathcal{P}$ is the lifting operator that maps the input to the first latent representation $z^{(0)}$, $\mathcal{L}^{(\ell)}$ is the $\ell$'th non-linear operator layer, and $\mathcal{Q}$ is the projection operator that maps the last latent representation $z^{(L)}$ to the output. On irregular geometries such as point clouds, we additionally define a coordinate map $\phi$, parameterized by a small neural network and learned end-to-end, that deforms the physical space of irregular geometry into a regular computational space. The architecture without this coordinate map is called FNO, while the one with the coordinate map is called geo-FNO. Fig. 2 (top) contains a schematic diagram of this iterative process.

Originally, Li et al. (2021a) formulate each operator layer as

$$\mathcal{L}^{(\ell)}\Big(z^{(\ell)}\Big) = \sigma\Big(W^{(\ell)} z^{(\ell)} + b^{(\ell)} + \mathcal{K}^{(\ell)}(z^{(\ell)})\Big), \tag{5}$$

where $\sigma : \mathbb{R} \to \mathbb{R}$ is a point-wise non-linear activation function, $W^{(\ell)} z^{(\ell)} + b^{(\ell)}$ is an affine point-wise map in the physical space, and $\mathcal{K}^{(\ell)}$ is a kernel integral operator using the Fourier transform,

$$\mathcal{K}^{(\ell)}\Big(z^{(\ell)}\Big) = \text{IFFT}\Big(R^{(\ell)} \cdot \text{FFT}(z)\Big) \tag{6}$$

The Fourier-domain weight matrices $\{R^{(\ell)} \mid \ell \in \{1, 2, \ldots, L\}\}$ take up most of the model size, requiring $O(LH^2M^D)$ parameters, where $H$ is the hidden size, $M$ is the number of top Fourier modes being kept, and $D$ is the problem dimension. Furthermore, the constant value for $M$ and the affine point-wise map allow the FNO to be resolution-independent.

**Our improved F-FNO architecture**  We propose changing the operator layer in Eq. (5) to:

$$\mathcal{L}^{(\ell)}\Big(z^{(\ell)}\Big) = z^{(\ell)} + \sigma\Big[W_2^{(\ell)} \sigma\big(W_1^{(\ell)} \mathcal{K}^{(\ell)}\big(z^{(\ell)}\big) + b_1^{(\ell)}\big) + b_2^{(\ell)}\Big] \tag{7}$$

Note that we apply the residual connection ($z^{(\ell)}$ term) *after* the non-linearity to preserve more of the layer input. We also use a two-layer feedforward, inspired by the feedforward design used in transformers (Vaswani et al., 2017). More importantly, we factorize the Fourier transforms over the problem dimensions, modifying Eq. (6) to

$$\mathcal{K}^{(\ell)}\Big(z^{(\ell)}\Big) = \sum_{d \in D} \Big[\text{IFFT}\Big(R_d^{(\ell)} \cdot \text{FFT}_d(z^{(\ell)})\Big)\Big] \tag{8}$$

Table 1: An overview of the datasets and the corresponding task.

| Dataset | Geometry | Dim. | Problem | Input | Output |
|---------|----------|------|---------|-------|--------|
| TorusLi | regular grid | 2D | Kolmogorov flow | $\omega_t$ | $\omega_{t+1}$ |
| TorusKochkov | regular grid | 2D | Kolmogorov flow | $\omega_t$ | $\omega_{t+1}$ |
| TorusVis | regular grid | 2D | Kolmogorov flow | $\omega_t$ and $\nu$ | $\omega_{t+1}$ |
| TorusVisForce | regular grid | 2D | Kolmogorov flow | $\omega_t$ and $\nu$ and $f_t$ | $\omega_{t+1}$ |
| Elasticity | point cloud | 2D | hyper-elastic material | point cloud | stress |
| Airfoil | structured mesh | 2D | transonic flow | mesh grid | velocity |
| Plasticity | structured mesh | 3D | plastic forging | boundary condition | displacement |

The seemingly small change from $R^{(\ell)}$ to $R_d^{(\ell)}$ in the Fourier operator reduces the number of parameters to $O(LH^2MD)$. This is particularly useful when solving higher-dimensional problems such as 3D plastic forging (Fig. 1d). The combination of the factorized transforms and residual connections allows the operator to converge in deep networks while continuing to improve performance (Fig. 3). It is also possible to share the weight matrices $R_d$ between the layers, which further reduces the parameters to $O(H^2MD)$. Fig. 2 (bottom) provides an overview of an F-FNO operator layer.

Furthermore, the F-FNO is highly flexible in its input representation, which means anything that is relevant to the evolution of the field can be an input, such as viscosity or external forcing functions for the torus. This flexibility also allows the F-FNO to be easily generalized to different PDEs.

**Training techniques to learn neural operators**   We find that a combination of deep learning techniques are very important for the FNO to perform well, most of which were overlooked in Li et al. (2021a)'s original implementation. The first is *enforcing the first-order Markov property*. We find Li et al. (2021a)'s use of the last 10 time steps as inputs to the neural operator to be unnecessary. Instead, it is sufficient to feed information only from the current step, just like a numerical solver. Unlike prior works (Li et al., 2021a; Kochkov et al., 2021), we do not unroll the model during training but instead use the *teacher forcing* technique which is often seen in time series and language modeling. In teacher forcing, we use the ground truth as the input to the neural operator. Finally during training, we find it useful to normalize the inputs and add a small amount of Gaussian noise, similar to how Sanchez-Gonzalez et al. (2020) train their graph networks. Coupled with cosine learning rate decay, we are able to make the training process of neural operators more stable. Ablation studies for the new representation and training techniques can be found in Fig. 3.

## 4   DATASETS AND EVALUATION SETTINGS

**PDEs on regular grids**   The four Torus datasets on regular grids (TorusLi, TorusKochkov, TorusVis, and TorusVisForce, summarized in Table 1) are simulations based on Kolmogorov flows which have been extensively studied in the literature (Chandler & Kerswell, 2013). In particular, they model turbulent flows on the surface of a 3D torus (i.e., a 2D grid with periodic boundary conditions). TorusLi is publicly released by Li et al. (2021a) and is used to benchmark our model against the original FNO. The ground truths are assumed to be simulations generated by the pseudo-spectral Crank-Nicholson second-order method on 64x64 grids. All trajectories have a constant viscosity $\nu = 10^{-5}$ (Re = 2000), use the same constant forcing function, $f(x, y) = 0.1[\sin(2\pi(x+y)) + \cos(2\pi(x+y))]$, and differ only in the initial field.

Using the same Crank-Nicolson numerical solver, we generate two further datasets, called TorusVis and TorusVisForce, to test the generalization of the F-FNO across Navier-Stokes tasks with different viscosities and forcing functions. In particular, for each trajectory, we vary the viscosity between $10^{-4}$ and $10^{-5}$, and set the forcing function to

$$f(t, x, y) = 0.1 \sum_{p=1}^{2} \sum_{i=0}^{1} \sum_{j=0}^{1} \left[ \alpha_{pij} \sin\left(2\pi p(ix + jy) + \delta t\right) + \beta_{pij} \cos\left(2\pi p(ix + jy) + \delta t\right) \right], \quad (9)$$

where the amplitudes $\alpha_{pij}$ and $\beta_{pij}$ are sampled from the standard uniform distribution. Furthermore, $\delta$ is set to 0 in TorusVis, making the forcing function constant across time; while it is set to 0.2 in TorusVisForce, giving us a time-varying force.

Finally, we regenerate TorusKochkov (Fig. 1a) using the same settings provided by Kochkov et al. (2021) but with different initial conditions from the original paper (since the authors did not release the full dataset). Here the ground truths are obtained from simulations on 2048x2048 grids using the pseudo-spectral Carpenter-Kennedy fourth-order method. The full-scale simulations are then downsampled to smaller grid sizes, allowing us to study the Pareto frontier of the speed vs accuracy space (see Fig. 4a). TorusKochkov uses a fixed viscosity of $0.001$ and a constant forcing function $\mathbf{f} = 4\cos(4y)\hat{x} - 0.1\mathbf{u}$, but on the bigger domain of $[0, 2\pi]$. Furthermore, we generate only 32 training trajectories to test how well the F-FNO can learn on a low-data regime.

**PDEs on irregular geometries** The Elasticity, Airfoil, and Plasticity datasets (final three rows in Table 1) are taken from Li et al. (2022). Elasticity is a point cloud dataset modeling the incompressible Rivlin-Saunders material (Pascon, 2019). Each sample is a unit cell with a void in the center of arbitrary shape (Fig. 1b). The task is to map each cloud point to its stress value. Airfoil models the transonic flow over an airfoil, shown as the white center in Fig. 1c. The neural operator would then learn to map each mesh location to its Mach number. Finally, Plasticity models the plastic forging problem, in which a die, parameterized by an arbitrary function and traveling at a constant speed, hits a block material from above (Fig. 1d). Here the task is to map the shape of the die to the $101 \times 31$ structured mesh over 20 time steps. Note that Plasticity expects a 3D output, with two spatial dimensions and one time dimension.

**Training details** For experiments involving the original FNO, FNO-TF (with teaching forcing), FNO-M (with the Markov assumption), and FNO-N (with improved residuals), we use the same training procedure as Li et al. (2021a). For our own models, we train for 100,000 steps on the regular grid datasets and for 200 epochs for the irregular geometry datasets, warming up the learning rate to $2.5 \times 10^{-3}$ for the first 500 steps and then decaying it using the cosine function (Loshchilov & Hutter, 2017). We use ReLU as our non-linear activation function, clip the gradient value at 0.1, and use the Adam optimizer (Kingma & Ba, 2015) with $\beta_1 = 0.9$, $\beta_2 = 0.999$, $\epsilon = 10^{-8}$. The weight decay factor is set to $10^{-4}$ and is decoupled from the learning rate (Loshchilov & Hutter, 2019). In each operator layer on the torus datasets, we always throw away half of the Fourier modes (e.g., on a 64x64 grid, we keep only the top 16 modes). Models are implemented in PyTorch (Paszke et al., 2017) and trained on a single Titan V GPU.

**Evaluation metrics** We use the normalized mean squared error as the loss function, defined as

$$\text{N-MSE} = \frac{1}{B} \sum_{i=1}^{B} \frac{\|\hat{\omega}_i - \omega\|_2}{\|\omega\|_2},$$

where $\|\cdot\|_2$ is the 2-norm, $B$ is the batch size, and $\hat{\omega}$ is the prediction of the ground truth $\omega$.

In addition to comparing the N-MSE directly, for TorusKochkov, we also compute the vorticity correlation, defined as

$$\rho(\omega, \hat{\omega}) = \sum_i \sum_j \frac{\omega_{ij}}{\|\omega\|_2} \frac{\hat{\omega}_{ij}}{\|\hat{\omega}\|_2},$$

and from which we measure the time until this correlation drops below 95%. To be consistent with prior work, we use the N-MSE to compare the F-FNO against the FNO and geo-FNO Li et al. (2021a; 2022), and the vorticity correlation to compare against Kochkov et al. (2021)'s work.

## 5 RESULTS FOR NAIVER-STOKES ON A TORUS

**Comparison against FNO** The performance on TorusLi is plotted in Fig. 3, with the raw numbers shown in Table A.3. We note that our method F-FNO is substantially more accurate than the FNO regardless of network depth, when judged by N-MSE. The F-FNO uses fewer parameters than the FNO, has a similar training time, but generally has a longer inference time. Even so, the inference time for the F-FNO is still up to two orders of magnitude shorter than for the Crank-Nicolson numerical solver.

In contrast to our method, Li et al. (2021a) do not use teacher forcing during training. Instead they use the previous 10 steps as input to predict the next 10 steps incrementally (by using each predicted

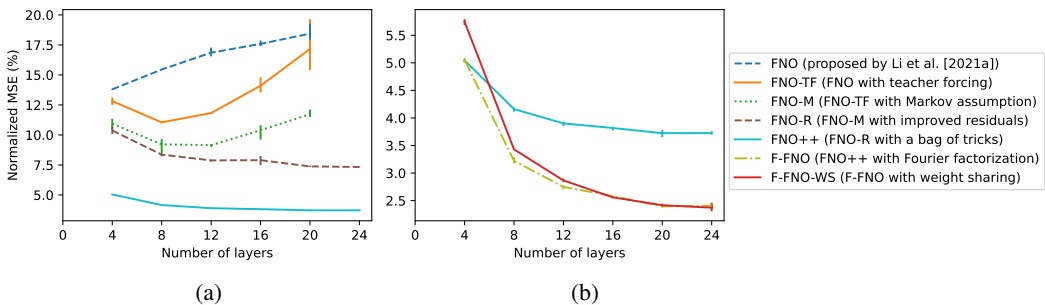

Figure 3: Performance (lower is better) on TorusLi, with error bars showing the min and max values over three trials. We show the original FNO (Li et al., 2021a), along with variants that use: teacher forcing, Markov assumption, improved residuals, a bag of tricks, Fourier factorization, and weight sharing. Note that F-FNO and F-FNO-WS are presented on a separate plot (b) to make visualizing the improvement easier (if shown in (a), F-FNO and F-FNO-WS would just be a straight line).

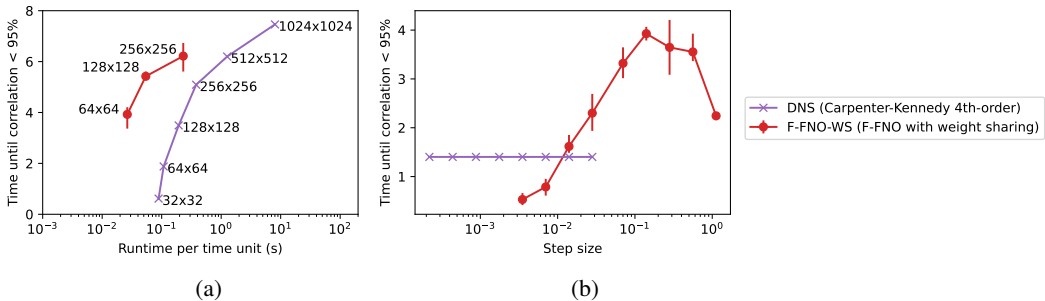

Figure 4: Performance of F-FNO on TorusKochkov. In (a), we plot the time until the correlation with the ground truths in the test set drops below 95% on the y-axis, against the time it takes to run one second of simulation on the x-axis. In (b), we show how, on the validation set of TorusKochkov, given a fixed spatial resolution of 64x64, changing the step size has no effect on the numerical solver; however there is an optimal step size for the F-FNO at around 0.2.

value as the input to the next step). We find that the teacher forcing strategy (FNO-TF, orange line), in which we always use the ground truth from the previous time step as input during training, leads to a smaller N-MSE when the number of layers is less than 24. Furthermore, enforcing the first-order Markov property (FNO-M, dotted green line), where only one step of history is used, further improves the performance over FNO and FNO-TF. Including two or more steps of history does not improve the results.

The models FNO, FNO-TF, and FNO-M do not scale with network depth, as seen by the increase in the N-MSE with network depth. These models even diverge during training when 24 layers are used. FNO-R, with the residual connections placed after the non-linearity, does not suffer from this problem and can finally converge at 24 layers. FNO++ further improves the performance, as a result of a careful combination of: normalizing the inputs, adding Gaussian noise to the training inputs, and using cosine learning rate decay. In particular, we find that adding a small amount of Gaussian noise to the normalized inputs helps to stabilize training. Without the noise, the validation loss at the early stage of training can explode.

Finally, if we use Fourier factorization (F-FNO, yellow dashed line), the error drops by an additional 35% ($3.73\% \rightarrow 2.41\%$) at 24 layers (Fig. 3b), while the parameter count is reduced by an order of magnitude. Sharing the weights in the Fourier domain (F-FNO-WS, red line) makes little difference to the performance especially at deep layers, but it does reduce the parameter count by another order of magnitude to 1M (see Fig. A.1 and Table A.3).

**Trade-off between speed and accuracy**    From Fig. 4a, we observe that our method F-FNO only needs 64x64 input grids to reach a similar performance to a 128x128 grid solved with DNS. At the

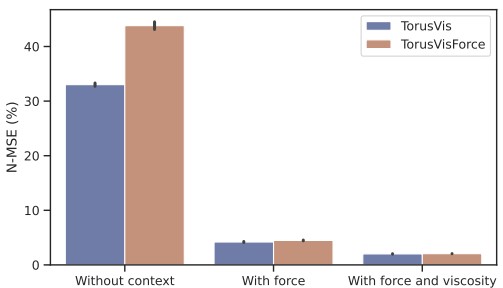
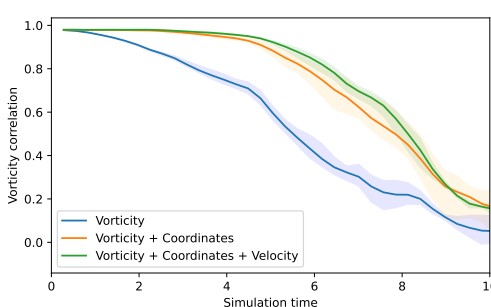

(a) Performance of F-FNO on different input features: having only vorticity as an input with no further context (first group); having vorticity and the force field as inputs (second group); and having vorticity, the force field, and viscosity as inputs (third group). The error bars are the standard deviation from three trials.

(b) Effect of having the coordinates and velocity as additional input channels on TorusKochkov. A higher line corresponds to a model that can correlate with the ground-truth vorticity for longer. Error bands correspond to min and max values from three trials.

Figure 5: Performance of F-FNO on different contexts and input representations.

same time, the F-FNO also achieves an order of magnitude speedup. While the highly specialized hybrid method introduced by Kochkov et al. (2021) can achieve a speedup closer to two orders of magnitude over DNS, the F-FNO takes a much more flexible approach and thus can be more easily adapted to other PDEs and geometries.

The improved accuracy of the F-FNO over DNS when both methods are using the same spatial resolution can be seen graphically in Fig. A.4. In this example, the F-FNO on a 128x128 grid produces a vorticity field that is visually closer to the ground truth than DNS running on the same grid size. This is also supported by comparing the time until correlation falls below $95\%$ in Fig. 4a.

**Optimal step size** The F-FNO includes a step size parameter which specifies how many seconds of simulation time one application of the operator will advance. A large step size sees the model predicting far into the future possibly leading to large errors, while a small step size means small errors have many more steps to compound. We thus try different step sizes in Fig. 4b.

In numerical solvers, there is a close relationship between the step size in the time dimension and the spatial resolution. Specifically, the Courant-Friedrichs-Lewy (CFL) condition provides an optimal step size given a space discretization: $\Delta t = C_{\max} \Delta x / \|\mathbf{u}\|_{\max}$. This means that if we double the grid size, the solver should take a step that is twice as small (we follow this approach to obtain DNS's Pareto frontier in Fig. 4a). Furthermore, having step sizes smaller than what is specified by the CFL condition would not provide any further benefit unless we also reduce the distance between two grid points (see purple line in Fig. 4b). On the other hand, a step size that is too big (e.g., bigger than 0.05 on a 64x64 grid) will lead to stability issues in the numerical solver.

For the F-FNO, we find that we can take a step size that is at least an order of magnitude bigger than the stable step size for a numerical solver. This is the key contribution to the efficiency of neural methods. Furthermore, there is a sweet spot for the step size – around 0.2 on TorusKochkov – and unlike its numerical counterpart, we find that there is no need to reduce the step size as we train the F-FNO on a higher spatial resolution.

**Flexible input representations** The F-FNO can be trained to handle Navier-Stokes equations with viscosities (in TorusVis) and time-varying forcing functions (in TorusVisForce) provided at inference time. Our model, when given both the force and viscosity, in addition to the vorticity, is able to achieve an error of 2% (Fig. 5a). If we remove the viscosity information, the error doubles. Removing the forcing function from the input further increases the error by an order of magnitude. This shows that the force has a substantial impact on the future vorticity field, and that the F-FNO can use information about the forcing function to make accurate predictions. More generally, different datasets benefit from having different input features – Table A.7 shows the minimum set of features to reach optimal performance on each of them. We also find that having redundant features does not significantly hurt the model, so there is no need to do aggressive feature pruning in practice.

Table 2: Performance (N-MSE, expressed as percentage, where lower is better) on point clouds (Elasticity) and structured meshes (Airfoil and Plasticity) between our F-FNO and the previous state-of-the-art geo-FNO (Li et al., 2022). Cells with a dash correspond to models which do not converge. The N-MSE is accompanied by the standard deviation from three trials. More detailed results are shown in Tables A.4 to A.6.

| No. of layers | Elasticity | | Airfoil | | Plasticity | |
|---|---|---|---|---|---|---|
| | geo-FNO | F-FNO | geo-FNO | F-FNO | geo-FNO | F-FNO |
| 4 layer | $2.5 \pm 0.1$ | $3.16 \pm 1.29$ | $1.9 \pm 0.4$ | $0.79 \pm 0.02$ | $0.74 \pm 0.01$ | $0.48 \pm 0.02$ |
| 8 layer | $3.3 \pm 1.3$ | $2.05 \pm 0.01$ | $1.4 \pm 0.5$ | $0.64 \pm 0.01$ | $0.57 \pm 0.04$ | $0.32 \pm 0.01$ |
| 12 layer | $16.8 \pm 0.7$ | $1.96 \pm 0.02$ | $4.1 \pm 4.4$ | $0.62 \pm 0.03$ | $0.45 \pm 0.03$ | $0.25 \pm 0.01$ |
| 16 layer | $16.3 \pm 0.4$ | $1.86 \pm 0.02$ | - | $0.61 \pm 0.01$ | - | $0.22 \pm 0.00$ |
| 20 layer | $16.0 \pm 0.7$ | $1.84 \pm 0.02$ | - | $0.57 \pm 0.01$ | - | $0.20 \pm 0.02$ |
| 24 layer | $15.9 \pm 0.5$ | $1.74 \pm 0.03$ | - | $0.58 \pm 0.04$ | - | $0.18 \pm 0.00$ |

Our experiments with different input representations also reveal an interesting performance gain from the *double encoding of information* (Fig. 5b). All datasets benefit from the coordinate encoding – i.e., having the $(x, y)$ coordinates as two additional input channels – even if the positional information is already contained in the absolute position of grid points (indices of the input array). We hypothesize that these two positional representations are used by different parts of the F-FNO. The Fourier transform uses the absolute position of the grid points and thus the Fourier layer should have no need for the $(x, y)$ positional features. However, the feedforward layer in the physical space is a pointwise operator and thus needs to rely on the raw coordinate values, since it would otherwise be independent of the absolute position of grid points.

## 6    RESULTS FOR PDEs ON POINT CLOUDS AND MESHES

As shown in Table 2, the geo-FNO (Li et al., 2022), similar to the original FNO, also suffers from network scaling. It appears to be stuck in a local minimum beyond 8 layers in the Elasticity problem and it completely fails to converge beyond 12 layers in Airfoil and Plasticity. Plasticity is the only task in which the geo-FNO gets better as we go from 4 to 12 layers ($0.74\% \rightarrow 0.45\%$). In addition to the poor scaling with network depth, we also find during our experiments that the geo-FNO can perform worse as we increase the hidden size $H$. This indicates that there might not be enough regularization in the model as we increase the model complexity.

Our F-FNO, on the other hand, continues to gain performance with deeper networks and bigger hidden size, reducing the prediction error by 31% on the Elasticity point clouds ($2.51\% \rightarrow 1.74\%$) and by 57% on the 2D transonic flow over airfoil problem ($1.35\% \rightarrow 0.58\%$). Our Fourier factorization particularly shines in the plastic forging problem, in which the neural operator needs to output a 3D array, i.e., the displacement of each point on a 2D mesh over 20 time steps. As shown in Table A.6, our 24-layer F-FNO with 11M parameters outperforms the 12-layer geo-FNO with 57M parameters by 60% ($0.45\% \rightarrow 0.18\%$).

## 7    CONCLUSION

The Fourier transform is a powerful tool to learn neural operators that can handle long-range spatial dependencies. By factorizing the transform, using better residual connections, and improving the training setup, our proposed F-FNO outperforms the state of the art on PDEs on a variety of geometries and domains. For future work, we are interested in examining equilibrium properties of generalized Fourier operators with an infinite number of layers and checking if the universal approximation property (Kovachki et al., 2021a) still holds under Fourier factorization.

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

# A    APPENDIX

Table A.1: An overview of the four fluid dynamics datasets on regular grids. Our newly generated datasets, TorusVis and TorusVisForce, contain simulation data with a more variety of viscosities and forces than TorusLi (Li et al., 2021a) and TorusKochkov (Kochkov et al., 2021). Note that Li et al. (2021a) did not generate a validation set.

| Dataset | Train / valid / test split | Trajectory length | Domain | Viscosity | Force varying across | |
|---|---|---|---|---|---|---|
| | | | | | samples | time |
| TorusLi | 1000 / 0 / 200 | 20 | $[0, 1]$ | $\nu = 10^{-5}$ | | |
| TorusKochkov | 32 / 4 / 4 | 34 | $[0, 2\pi]$ | $\nu = 10^{-3}$ | | |
| TorusVis | 1000 / 200 / 200 | 20 | $[0, 1]$ | $\nu \in [10^{-5}, 10^{-4})$ | ✓ | |
| TorusVisForce | 1000 / 200 / 200 | 20 | $[0, 1]$ | $\nu \in [10^{-5}, 10^{-4})$ | ✓ | ✓ |

Table A.2: An overview of the three PDE datasets on irregular geometries. These datasets were generated by Li et al. (2022).

| Dataset | Train | Valid | Test | Governing equation | Problem dimension |
|---|---|---|---|---|---|
| Elasticity | 1000 | 200 | 200 | Equation of a solid body | point cloud on 2D unit cell |
| Airfoil | 1000 | 200 | 200 | Euler's equation | 2D structured mesh |
| Plasticity | 827 | 80 | 80 | Equation of a solid body | 2D structured mesh over 1D time |

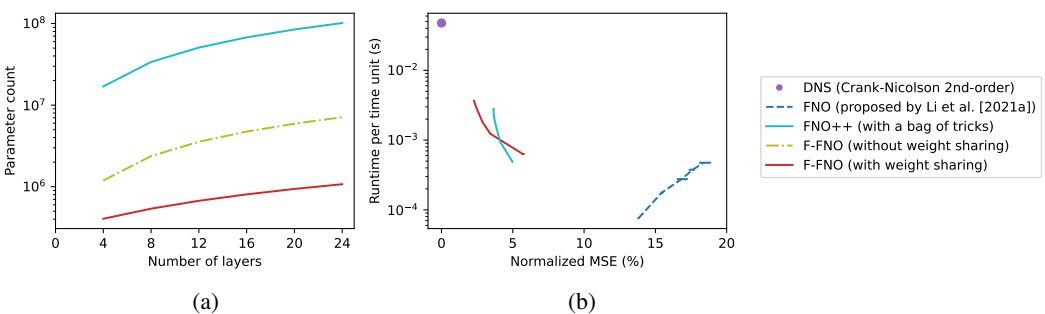

(a)                    (b)

Figure A.1: The resource usage of four model variants, in terms of (a) the parameter count and (b) inference time (the time it takes to run one second of simulation). Error bars, when applicable, show the min and max values over three trials. In (b), as we move along a line, we increase the number of layers. We observe that only our model variants (F-FNO) have the desired slope, that is, as we use more resources (increasing the inference time), we obtain better predictions.

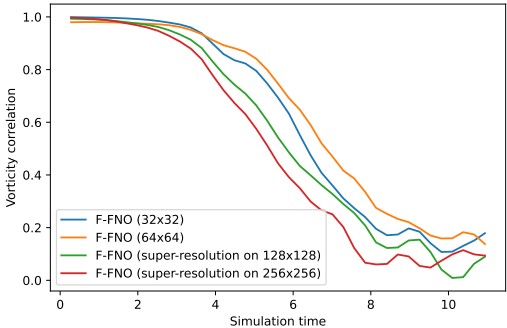 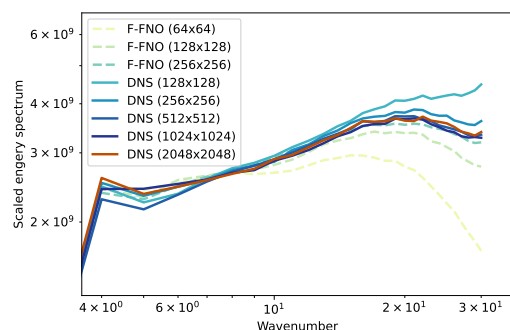

(a) Zero-shot super-resolution performance of F-FNO. We train the model on 32x32 and 64x64 grids of TorusKochkov, and evaluate on the larger 128x128 and 256x256 grids. We observe some degradation in the correlation with the ground truths on unseen grid sizes.

(b) Energy spectra of F-FNO and DNS on various grid sizes. The spectra are computed by averaging the kinetic energy for each wavenumber between $t = 12$ and $t = 34$, when the predictions from all methods have decorrelated with the ground truths.

Figure A.2: Performance of F-FNO on zero-shot superresolution and its ability to capture the energy spectrum of DNS on TorusKochkov.

**Zero-shot super-resolution**    In Fig. A.2a, we train the F-FNO once on 32x32 and 64x64 grids from TorusKochkov, and then perform inference and evaluation on 128x128 and 256x256 grids. This extends the super-resolution setting presented by Li et al. (2021a) as they only worked on simple PDEs such as the 1D Burger's equation and the 2D Darcy flow. We find that although the F-FNO can do zero-shot super-resolution – unlike a traditional CNN which by design cannot even accept inputs of variable size – its performance does degrade on grid sizes not seen during training. This is seen by the lower vorticity correlation of the super-resolution F-FNO settings in Fig. A.2a. We posit that the super-resolution performance could be improved by training on a variety of grid sizes (e.g., by downsampling each training example to a random size). We leave such exploration for future work.

**Capturing the energy spectrum**    In addition to having a high vorticity correlation, a good model should also produce predictions with an energy spectrum similar to the most accurate numerical methods. Given the Fourier transform of a velocity field $\hat{\mathbf{u}} = \text{FFT}(\mathbf{u})$, we can compute, for each wavenumber $k$, the corresponding kinetic energy as $E(k) = \frac{1}{2}\|\hat{\mathbf{u}}_k\|^2$. Fig. A.2b shows the energy spectrum of both the F-FNO and DNS at different resolutions. These multiple DNS resolutions are included both as a reference solution in the case of DNS 2048x2048, and to demonstrate that increasing the resolution of DNS further is not likely to substantially change the energy spectrum.

We observe that compared to DNS on 2048x2048, the F-FNO trained on 64x64 grids produces an energy spectrum that has substantially lower energy at high wavenumbers. This is expected as at this spatial resolution we only select the top 16 modes in each Fourier layer. Even so, the F-FNO can still capture the long term trend much better than running DNS on a grid four times its size (see Fig. 4a). As we select more Fourier modes on bigger grids (top 32 modes on 128x128 grids and top 64 modes on 256x256 grids), the energy spectrum produced converges towards that of the reference solution (DNS on 2048x2048). This gives some indication that the F-FNO is able to accurately predict both high and low frequency details.

**Effect of using cosine transforms**    As an alternative to the Fourier transform, Poli et al. (2022) proposed using the cosine transform, which has the advantage of being real-valued, thus halving the number of parameters. Let the Factorized Cosine Neural Operator (F-CNO) be the operator where the Fourier transform is replaced with the cosine transform. In Fig. A.3, we observe that on Airfoil, the F-CNO outperforms the F-FNO especially at deeper layers. On Plasticity, the F-CNO performs comparably to the F-FNO on the same depth, while using fewer parameters. We have not had much success in training the F-CNO on torus datasets such as TorusKochkov. We leave the investigation of how stable the cosine transform is on different domains to future work.

Table A.3: Detailed performance on TorusLi. These results are used to generate Fig. 3 in the main paper. We run three trials for each experiment, each with a different random seed. We report the mean N-MSE from the three trials, along with the min and max value. A dash indicates that the data is not available.

| | No. of layers | No. of parameters | N-MSE (%) | | | Training time (h) |
|---|---|---|---|---|---|---|
| | | | Mean | Min | Max | |
| ResNet (Li et al., 2021a) | - | 266,641 | 27.53 | - | - | |
| TF-Net (Li et al., 2021a) | - | 7,451,724 | 22.68 | - | - | |
| U-Net (Li et al., 2021a) | - | 7,451,724 | 19.82 | - | - | |
| FNO (Li et al., 2021a) | 4 | 414,517 | 15.56 | - | - | |
| FNO (reproduced) | 4 | 926,357 | 13.80 | 13.75 | 13.83 | 2 |
| | 8 | 1,849,637 | 15.45 | 15.40 | 15.50 | 3 |
| | 12 | 2,772,917 | 16.86 | 16.55 | 17.27 | 5 |
| | 16 | 3,696,197 | 17.59 | 17.41 | 17.87 | 6 |
| | 20 | 4,619,477 | 18.44 | 17.91 | 19.24 | 7 |
| | 24 | | does not converge | | | |
| FNO-TF (FNO with teacher forcing) | 4 | 926,357 | 12.82 | 12.50 | 13.13 | 2 |
| | 8 | 1,849,637 | 11.05 | 10.98 | 11.10 | 3 |
| | 12 | 2,772,917 | 11.83 | 11.74 | 11.91 | 5 |
| | 16 | 3,696,197 | 14.10 | 13.55 | 14.80 | 6 |
| | 20 | 4,619,477 | 17.17 | 15.40 | 19.64 | 7 |
| | 24 | | does not converge | | | |
| FNO-M (FNO-TF with Markov assumption) | 4 | 926,177 | 10.94 | 10.47 | 11.35 | 1 |
| | 8 | 1,849,457 | 9.22 | 8.48 | 9.67 | 2 |
| | 12 | 2,772,737 | 9.15 | 9.02 | 9.25 | 3 |
| | 16 | 3,696,017 | 10.39 | 9.61 | 10.81 | 4 |
| | 20 | 4,619,297 | 11.73 | 11.50 | 12.12 | 4 |
| | 24 | | does not converge | | | |
| FNO-R (FNO-M with improved residuals) | 4 | 926,177 | 10.37 | 10.08 | 10.69 | 1 |
| | 8 | 1,849,457 | 8.36 | 8.22 | 8.46 | 2 |
| | 12 | 2,772,737 | 7.88 | 7.78 | 7.95 | 3 |
| | 16 | 3,696,017 | 7.90 | 7.48 | 8.25 | 4 |
| | 20 | 4,619,297 | 7.38 | 7.34 | 7.45 | 5 |
| | 24 | 5,542,577 | 7.33 | 7.31 | 7.36 | 5 |
| FNO++ (FNO-R with bags of tricks) | 4 | 16,919,746 | 5.05 | 5.02 | 5.06 | 1 |
| | 8 | 33,830,594 | 4.16 | 4.11 | 4.19 | 2 |
| | 12 | 50,741,442 | 3.90 | 3.85 | 3.93 | 3 |
| | 16 | 67,652,290 | 3.82 | 3.77 | 3.84 | 4 |
| | 20 | 84,563,138 | 3.72 | 3.65 | 3.79 | 5 |
| | 24 | 101,473,986 | 3.73 | 3.70 | 3.76 | 5 |
| F-FNO (FNO++ with Fourier factorization) | 4 | 1,191,106 | 5.05 | 5.00 | 5.09 | 1 |
| | 8 | 2,373,314 | 3.22 | 3.17 | 3.29 | 2 |
| | 12 | 3,555,522 | 2.75 | 2.70 | 2.77 | 3 |
| | 16 | 4,737,730 | 2.58 | 2.57 | 2.59 | 4 |
| | 20 | 5,919,938 | 2.39 | 2.37 | 2.42 | 5 |
| | 24 | 7,102,146 | 2.41 | 2.37 | 2.47 | 6 |
| F-FNO-WS (F-FNO with weight sharing) | 4 | 404,674 | 5.74 | 5.69 | 5.79 | 1 |
| | 8 | 538,306 | 3.43 | 3.42 | 3.44 | 2 |
| | 12 | 671,938 | 2.87 | 2.84 | 2.90 | 3 |
| | 16 | 805,570 | 2.56 | 2.54 | 2.57 | 4 |
| | 20 | 939,202 | 2.42 | 2.38 | 2.45 | 5 |
| | 24 | 1,072,834 | 2.37 | 2.31 | 2.45 | 6 |

Table A.4: Detailed performance on Airfoil. These results are more detailed version of Table 2 in the main paper. We run three trials for each experiment, each with a different random seed. We report the mean N-MSE from the three trials, along with the min and max value.

| | No. of layers | No. of parameters | N-MSE (%) | | | Training time (h) |
|---|---|---|---|---|---|---|
| | | | Mean | Min | Max | |
| geo-FNO (reproduced) | 4 | 2,368,033 | 1.87 | 1.40 | 2.27 | 4 |
| | 8 | 4,731,553 | 1.35 | 1.02 | 2.00 | 5 |
| | 12 | 7,095,073 | 4.11 | 0.92 | 10.29 | 4 |
| F-FNO | 4 | 1,715,458 | 0.79 | 0.76 | 0.82 | 3 |
| | 8 | 3,421,954 | 0.64 | 0.63 | 0.65 | 4 |
| | 12 | 5,128,450 | 0.62 | 0.59 | 0.67 | 5 |
| | 16 | 6,834,946 | 0.61 | 0.59 | 0.62 | 5 |
| | 20 | 8,541,442 | 0.57 | 0.56 | 0.58 | 4 |
| | 24 | 10,247,938 | 0.58 | 0.56 | 0.64 | 4 |
| F-FNO-WS (F-FNO with weight sharing) | 4 | 535,810 | 0.98 | 0.90 | 1.03 | 0.4 |
| | 8 | 669,442 | 0.72 | 0.70 | 0.75 | 0.7 |
| | 12 | 803,074 | 0.68 | 0.66 | 0.70 | 1 |
| | 16 | 936,706 | 0.67 | 0.63 | 0.70 | 1 |
| | 20 | 1,070,338 | 0.64 | 0.63 | 0.66 | 2 |
| | 24 | 1,203,970 | 0.66 | 0.60 | 0.70 | 2 |

Table A.5: Detailed performance on Elasticity. These results are more detailed version of Table 2 in the main paper. We run three trials for each experiment, each with a different random seed. We report the mean N-MSE from the three trials, along with the min and max value. Note that for a given layer, our F-FNO (whether with weight sharing or without) has slightly more parameters than the geo-FNO. This is due to the F-FNO using a bigger hidden size $H$. We find that on the geo-FNO, increasing its hidden size does not necessarily translate to a better performance.

| | No. of layers | No. of parameters | N-MSE (%) | | | Training time (h) |
|---|---|---|---|---|---|---|
| | | | Mean | Min | Max | |
| geo-FNO (reproduced) | 4 | 1,546,403 | 2.51 | 2.43 | 2.59 | 0.4 |
| | 8 | 2,730,659 | 3.30 | 2.32 | 5.10 | 0.5 |
| | 12 | 3,914,915 | 16.76 | 16.17 | 17.72 | 0.7 |
| F-FNO | 4 | 3,205,763 | 3.16 | 2.23 | 4.98 | 1 |
| | 8 | 4,338,051 | 2.05 | 2.04 | 2.06 | 1 |
| | 12 | 5,470,339 | 1.96 | 1.93 | 1.98 | 2 |
| | 16 | 6,602,627 | 1.86 | 1.83 | 1.88 | 2 |
| | 20 | 7,734,915 | 1.84 | 1.82 | 1.86 | 2 |
| | 24 | 8,867,203 | 1.74 | 1.70 | 1.78 | 2 |
| F-FNO-WS (F-FNO with weight sharing) | 4 | 2,681,475 | 3.55 | 2.36 | 5.84 | 0.2 |
| | 8 | 2,765,187 | 2.23 | 2.18 | 2.29 | 0.2 |
| | 12 | 2,848,899 | 2.12 | 2.10 | 2.16 | 0.3 |
| | 16 | 2,932,611 | 2.08 | 2.06 | 2.10 | 0.4 |
| | 20 | 3,016,323 | 2.04 | 1.99 | 2.07 | 0.5 |
| | 24 | 3,100,035 | 1.97 | 1.94 | 2.01 | 0.6 |

Table A.6: Detailed performance on Plasticity. These results are more detailed version of Table 2 in the main paper. We run three trials for each experiment, each with a different random seed. We report the mean N-MSE from the three trials, along with the min and max value.

| | No. of layers | No. of parameters | N-MSE (%) | | | Training time (h) |
| --- | --- | --- | --- | --- | --- | --- |
| | | | Mean | Min | Max | |
| geo-FNO (reproduced) | 4 | 18,883,492 | 0.74 | 0.73 | 0.75 | 2 |
| | 8 | 37,762,084 | 0.57 | 0.55 | 0.63 | 4 |
| | 12 | 56,640,676 | 0.45 | 0.41 | 0.49 | 5 |
| F-FNO | 4 | 1,846,920 | 0.48 | 0.47 | 0.51 | 4 |
| | 8 | 3,684,488 | 0.32 | 0.31 | 0.34 | 8 |
| | 12 | 5,522,056 | 0.25 | 0.24 | 0.26 | 12 |
| | 16 | 7,359,624 | 0.22 | 0.21 | 0.22 | 16 |
| | 20 | 9,197,192 | 0.20 | 0.18 | 0.22 | 20 |
| | 24 | 11,034,760 | 0.18 | 0.17 | 0.18 | 24 |
| F-FNO-WS (F-FNO with weight sharing) | 4 | 568,968 | 0.58 | 0.57 | 0.60 | 4 |
| | 8 | 702,600 | 0.50 | 0.46 | 0.52 | 8 |
| | 12 | 836,232 | 0.44 | 0.42 | 0.48 | 12 |
| | 16 | 969,864 | 0.40 | 0.36 | 0.44 | 16 |
| | 20 | 1,103,496 | 0.34 | 0.31 | 0.37 | 19 |
| | 24 | 1,237,128 | 0.30 | 0.28 | 0.35 | 21 |

Table A.7: The F-FNO is flexible in its input representation. We find that different datasets benefit from having different features. Shown here is the optimal input combination for each dataset on the torus.

| Dataset | Vorticity | Velocity | Coordinates | Viscosity | Forcing |
| --- | --- | --- | --- | --- | --- |
| TorusLi | ✓ | | ✓ | | |
| TorusKochkov | ✓ | ✓ | ✓ | | |
| TorusVis | ✓ | | ✓ | ✓ | |
| TorusVisForce | ✓ | | ✓ | ✓ | ✓ |

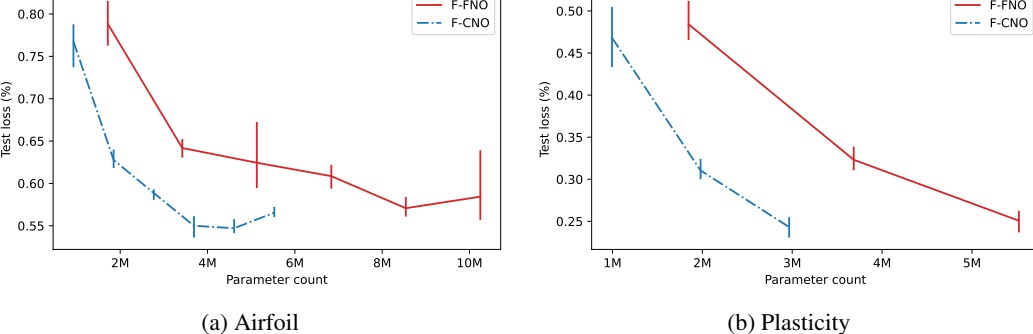

(a) Airfoil  (b) Plasticity

Figure A.3: Effect of the cosine transform on Airfoil and Plasticity. We plot the test loss (y-axis) against the model parameter count (x-axis). Error bars show the min-max values from three trials. As we move a long each line, we make the network deeper, which increases the number of parameters. On Airfoil (a), the F-CNO outperforms the F-FNO at deeper layers. On Plasticity (b), the performance between the two is mostly similar for the same depth. Since cosine transforms are real-valued, the F-CNO requires only half as many parameters as the F-FNO.

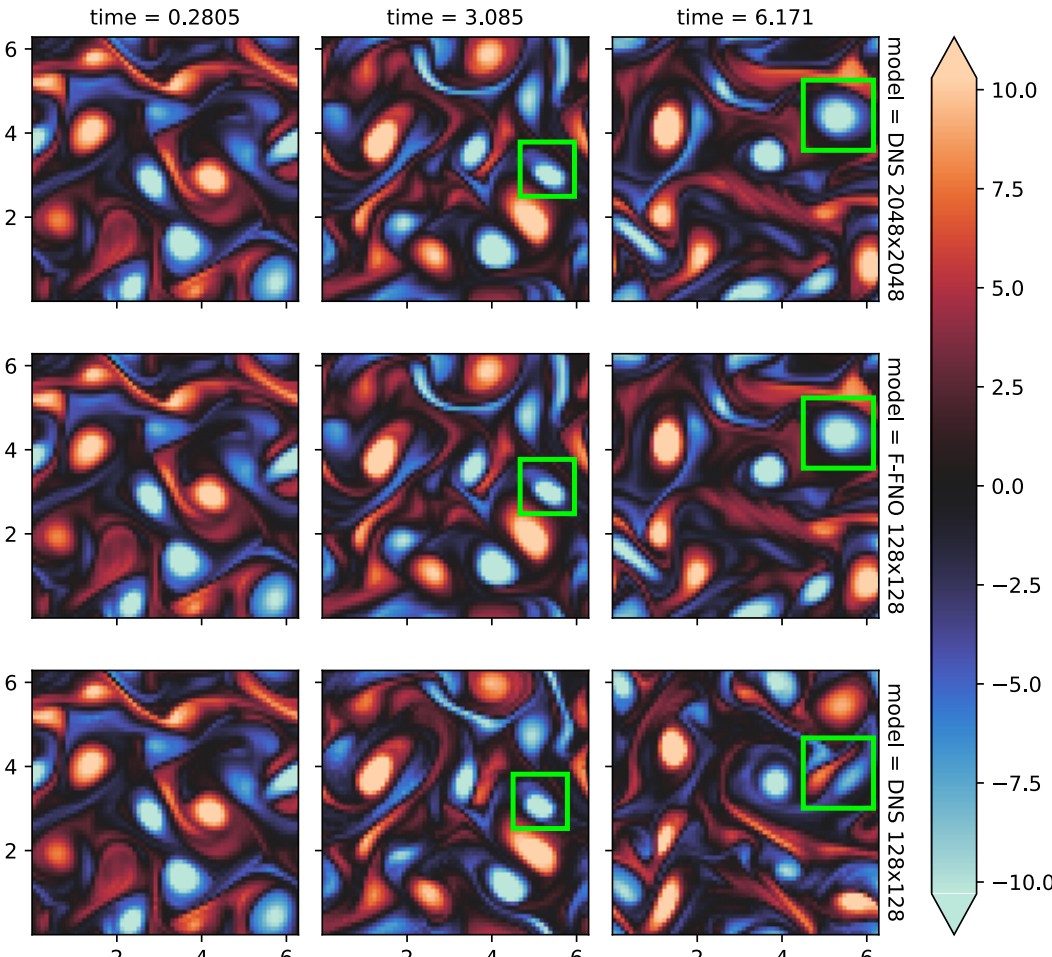

Figure A.4: Similar to Kochkov et al. (2021), we visualize how the correlation with the ground truths varies between different models. The heatmaps represent the surface of a torus mapped onto a 2D grid, with color representing the vorticity (the spinning motion) of the fluid. We observe that the vorticity fields predicted by the F-FNO trained on 128x128 grids (middle row) correlates with the ground truths (top row) for longer than if we run DNS on the same spatial resolution (bottom row). This is especially evident after 6 seconds of simulation time (compare the green boxes). In other words, for the same desired accuracy, the F-FNO requires a smaller grid input than a numerical solver. This observation is also backed up by Fig. 4a.

