# OpenReview forum: "Factorized Fourier Neural Operators"
_ICLR.cc/2023/Conference — ICLR 2023 poster_

### Official Review · Reviewer_jqw2 · 2022-10-14

**Confidence:** 3
**Correctness:** 3
**Technical Novelty And Significance:** 1
**Empirical Novelty And Significance:** 2
**Recommendation:** 6

**Clarity, Quality, Novelty And Reproducibility:**

The clarity is OK but the writing is poor. This means although the authors presented what they have done, they did not bridge the findings to their central claims well. Moreover, the contributions are not well presented.

The quality is incremental and I think the most valuable part of this work has not been presented well through the current draft.

Not too much originality can be found in this work.

**Strength And Weaknesses:**

# Strengths
1. The results seem to be impressive.
2. This paper clearly presented their method.
3. Some of the visualizations might inspire other researchers in this domain.

# Weaknesses

## 1. The proposed contributions are incremental and are not even well-defined.
Supporting details:

(1) The residual connections are not novel, and a naive application is not enough for ICLR.

(2) In the abstract, the authors say, "carefully designed training strategies." This claim is too vague.

(3) In the introduction, the authors claim, "In our exploration of various regularization techniques, we find that it is possible to reduce the parameter count by up to an order of magnitude while still outperforming state of the art by a significant margin." This is too incremental. Applying a kind of regularization cannot be a major contribution. Also, how and why the regularization work remains unclear.

(4) The core of the introduction and the method are too few. The major body of the methods is about the formulation of the problem and the baseline methods.

(5) In the introduction, the authors claim three key contributions. The first is about the factorized Fourier neural operator. However, why is it called factorized? I think the only change is the residual connection. This should not be called the factorized Fourier neural operator, and this contribution is still weak.

(6) In the introduction, the authors list two of the experimental findings as their contributions. However, the experimental findings are not that important because the technical contributions are weak, and the experiments are not convincing.

## 2. The experimental results are not convincing. In general, I do not believe the baselines and the improved models are really working well.

(1)  All the error numbers are too large. F-FNO achieves 3.16 errors in elasticity. However, the original FNO paper reports numbers at ~0.01. I understand the settings might be different. If this is the case, why do not the authors put all the methods under the same setting?

(2) Can the authors provide a visualization comparing the GT with the prediction? Why does the modification work under this setting?

(3) Is the baseline strong? Why don't the authors conduct experiments on the official settings of FNO?

(4) In figure captions, the briefing settings should be presented instead of saying, "our models work well, ... ". It is always better for the readers to discover the improvements from the figures rather than reading general over-stated claims.

(5) In Figure 2, how to understand the figure? What are the inputs to each model? What are the outputs? Why are there big holes in the figure? What does the color mean? How to understand the differences between different modalities? Can the baselines achieve this? If the baselines cannot support varying geometries, why? What are the corresponding gt? What do geometries mean here? Are the geometries important in this domain?

(6) The error bars should be shown in each figure.

(7) It seems that the changes are not significant. The improvements should not be so significant.



**Summary Of The Paper:**

This paper presents the factorized Fourier neural operator (F-FNO), which brings a set of techniques on FNO. The first technique is the separable Fourier representation to improve model stability and performance. Second, the improved residual connections. Third, the training strategies are carefully designed. Impressive results are found on a variety of benchmarks.

**Summary Of The Review:**

Incremental contributions; Clearly below the acceptance bar; results are not convincing.

——————————————————————————————————————————————————————
After discussing with other reviewers via the Zoom meeting, some of my questions are addressed, and it would be better if the authors clarify the mentioned points in the next revision. Also, some senior reviewers with great expertise in this domain insist that this paper has a clear empirical advantage over the baselines. Therefore, I agree to accept this paper right now. I don't think a simple method should not be accepted, whereas the AC also thinks so. I think the authors should clarify their central contribution (eq.8) in this case and not hide their contribution behind a set of tricks. I think a revision would be beneficial to the community and also to the authors.

---

> ### Author Response · Authors · 2022-11-19
> **Response to Reviewer jqw2 Part 1**
>
> Thank you for the comments and questions.
>
> > (1.1) The residual connections are not novel.
>
> Although residual connections are not new, where to place them in the neural network is tricky. The original FNO put the residual connection before the activation layer. We show that placing it after activation makes a big impact (reducing error from 9.2% to 7.3%), and finally allowing the model to reduce (rather than increase) the error as we add more layers (see revised Figure 3, brown-dashed, the second line from the bottom).
>
> > (1.2) In the abstract "carefully designed training strategies" is vague.
>
> We have rephrased the corresponding sentence in the abstract to *“training strategies such as the Markov assumption, Gaussian noise, and cosine learning rate”* (also see key changes 1 and 3 in our Summary Response). This claim is backed up in Section 5.
>
> > (1.3) In the introduction, the authors claim, "In our exploration of various regularization techniques…" This is too incremental. Applying a kind of regularization cannot be a major contribution. Also, how and why the regularization work remains unclear.
>
> We have now removed this phrase and rewrote the Introduction. Please see our updated contribution list. Specifically:
> 1) *We propose a new representation, the F-FNO, which consists of separable Fourier representation and improved residual connections, reducing the model complexity and allowing it to scale to deeper networks (Fig. 2 and Eqs. (7) and (8)).*
> 2) *We improve the learning process of neural operators by incorporating training techniques such as Markov assumption, Gaussian noise, and cosine learning rate decay (Fig. 3); and investigate how well the operator can handle different input representations (Fig. 5).*
>
> > (1.4) The core of the introduction and the method are too few. The major body of the methods is about the formulation of the problem and the baseline methods.
>
> We rewrote the Introduction to clearly illustrate the task of solving PDEs, the shortcomings of the first generation of neural operators, and our contributions. We have also rewritten Section 3 to clearly mark our novel contributions (under “Our improved F-FNO architecture” heading and, in particular, Equations 7 and 8).
>
> > (1.5) In the introduction, the authors claim three key contributions. The first is about the factorized Fourier neural operator. However, why is it called factorized? I think the only change is the residual connection. This should not be called the factorized Fourier neural operator, and this contribution is still weak.
>
> We’d like to clarify that factorization and residuals are two separate contributions.
>
> Fourier Factorization is shown in Equation 8 and illustrated in Figure 1. Here factorization means that we compute the Fourier transform on each dimension independently – there are no interaction terms between the dimensions when we are in the Fourier space. The factorized structure reduces the number of parameters from $O(LH^2M^D)$  to $O(LH^2MD)$ parameters, where $D$ is the number of dimensions. This is particularly useful when solving higher-dimensional problems (e.g., 3D Plasticity). Table A.6 backs up this claim on reducing model complexity in 3D.
>
> > (1.6) In the introduction, the authors list two of the experimental findings as their contributions. However, the experimental findings are not that important because the technical contributions are weak, and the experiments are not convincing.
>
> Thank you for this comment. We have clarified our contributions to representation, training strategies, and empirical performance in key change 1 in our Summary Repsonse. Also see the reply to (1.1), (1.3), and (1.5) above.
>
> On the experimental findings, please see our response to (2.1).

---

> ### Author Response · Authors · 2022-11-19
> **Response to Reviewer jqw2 Part 2**
>
> > (2.1) All the error numbers are too large
>
> We’d like to clarify an important misunderstanding of the evaluation metric: we report F-FNO (and comparator) errors as percentages, while the original FNO paper [Li2021FNO] reports the normalized mean-squared error (N-MSE) as decimal. We use percentages to avoid showing too many leading zeros and improve readability.
>
> For example, the baseline geo-FNO in [Li2022GeoFNO] achieves an N-MSE of 0.0229 (or 2.29%) on the Elasticity data. In Table 2 of our paper, our reproduced geo-FNO achieves 0.025 +- 0.001 (or 2.5% +- 0.1%). There’s still a small discrepancy - we posit three reasons for this: random seed for weight initialization and training batches, software package versions (not provided on the code release), and the fact that  [Li2022GeoFNO] did not report error bars. The remaining settings are exactly the same: hyper-parameters, batch size, optimizer, etc.
>
> We have clarified the unit of N-MSE in Table 2, verified that the unit of N-MSE has been specified in all other Figures and Tables, and ensured that the unit is always present when discussing the results in Section 6.
>
> > (2.2) visualization comparing the GT with the prediction
>
> See Figure A.4 in the revised submission (Fig A.3 in the original)  for a comparison of the F-FNO predictions with ground truths on the TorusKochkov dataset.
>
> For the irregular geometry datasets (e.g., Elasticity, Plasticity, Airfoil) the difference between F-FNO prediction and ground truths is not visible to the naked eye, since most errors are 1% or less.
>
> > (2.3) Why don't the authors conduct experiments on the official settings of FNO?
>
> See our responses to (2.1) and (2.7).
>
> > (2.4) In figure captions, the briefing settings should be presented
>
> Thank you for the suggestion. Figure and table captions have been reworded to contain only the settings info.
>
> > (2.5.1) In Figure 2, how to understand the figure? What are the inputs to each model? What are the outputs? Why are there big holes in the figure? What does the color mean?
>
> We have updated Figure 1 in the revised paper (Figure 2 in the original) to clearly illustrate the input and output of each dataset. Corresponding governing equations have now been added to Section 3.
>
> On the Elasticity data, we can think of the hole in the middle as a boundary condition that applies stress on the material. The goal is to predict the stress value (represented by the color, redder means higher stress) of each point on the cloud point.
>
> On Airfoil, the white hole in the middle represents the cross-section of an aircraft wing, and the mesh surrounding it models the flow velocity (represented by the color) of air around the wing at equilibrium.
>
> We refer the readers to [Li2022GeoFNO] for a more complete description of the datasets, including how the training samples are generated. Our paper can only provide a brief overview due to the page limit.
>
> > (2.5.2) How to understand the differences between different modalities? Can the baselines achieve this? If the baselines cannot support varying geometries, why? What are the corresponding gt? What do geometries mean here? Are the geometries important in this domain?
>
> Geometry refers to the shape of the domain. Examples of geometries include a regular grid (revised Figure 1a), a point cloud (revised Figure 1b), and a structured mesh (revised Figure 1c). Geometries are important because in many problems, we require higher resolutions in some parts of the domains than others. For example, when modeling the airflow around an aircraft wing, we need a higher resolution near the tip of the airfoil (see Figure 1c), hence a mesh is a more appropriate geometry than a grid.
>
> The ground truths are generated using traditional numerical methods. Please see the updated Figure 1 and the output column in Table 1 in the revised paper for what we are trying to predict in each dataset.
>
> The baselines are the FNO [Li2021FNO] and geo-FNO [Li2022GeoFNO]. Please note one possible point of confusion. When [Li2021FNO] proposed the FNO, it only worked with regular grids because the key component (the discrete Fourier transform) can only be applied to a regular grid. Later [Li2022GeoFNO] adds an extra coordinate map to the FNO and call the resulting model geo-FNO. This coordinate map deforms an irregular real space into a regular computational space, thus allowing us to learn on irregular geometries such as point clouds. The rest of the geo-FNO architecture is identical to that of the FNO. We have updated Section 2 to clarify this point.
>
> Our F-FNO can work both without the coordinate map on regular grids (Figure 3 in the revised paper), and with the coordinate map on irregular geometries (Table 2).

---

> > ### Author Response · Authors · 2022-11-19
> > **Response to Reviewer jqw2 Part 2 (continued)**
> >
> > > (2.6) Error bars
> >
> > Figure 3 and Figure 5a already have error bars, some of which are too small to see. Figure 4 and 5b has now been updated with error bars and error bands, respectively. Some baselines, such as the Direct Numerical Simulation (DNS) baseline, do not have error bars because there is no random component/learned weight in the solver.
> >
> > > (2.7) Changes are not significant
> >
> > Given our clarification in (2.1), we’d like to re-iterate the overall performance. On the same datasets and under the same settings as [Li2021FNO] and [Li2022GeoFNO], the F-FNO outperforms the state of the art on TorusLi (13.8% → 2.4%), Elasticity (2.5% → 1.74%), Airfoil (1.4% → 0.58%), and Plasticity (0.45% → 0.18%). All of these reductions are statistically significant as indicated by the error bars in Figure 3 and Table 2. The Appendix contains more detailed results, showing the N-MSE min-max range of each experiment.
> >
> > See Figure 3 (backed up by Table A.3 and discussed in detail in Section 5) for a detailed breakdown of all the contributions. See our response to (1.1) for a summary of this breakdown.
> >
> > Anonymized code is attached in the supplementary zip file. We will release all pre-trained model weights and publish the code on GitHub upon paper publication for reproducibility.
> >
> > **References**
> >
> > [Li2021FNO] Zongyi Li, Nikola B. Kovachki, K. Azizzadenesheli, Burigede Liu, K. Bhattacharya, Andrew Stuart, and Anima Anandkumar. Fourier neural operator for parametric partial differential equations. In International Conference on Learning Representations, 2021a. URL https://openreview.net/forum?id=c8P9NQVtmnO.
> >
> > [Li2022GeoFNO] Zong-Yi Li, Daniel Z. Huang, Burigede Liu, and Anima Anandkumar. Fourier neural operator with learned deformations for pdes on general geometries. ArXiv, abs/2207.05209, 2022.
> >
> > [Kovachki2021] Nikola Kovachki, Samuel Lanthaler, and Siddhartha Mishra. On universal approximation and error bounds for fourier neural operators. Journal of Machine Learning Research, 22(290):1–76, 2021a. URL http://jmlr.org/papers/v22/21-0806.html.

---

> > > ### Comment · Reviewer_jqw2 · 2022-11-19
> > > **Thanks for the response.**
> > >
> > > I think some of the issues are partially resolved, but not very perfectly.
> > > 1. About the contribution. The authors re-write their contributions, and I think the paper is worth another review. The author tries to state they make several contributions in this domain, where the contributions are made from several training tricks and pieces of model components. I suggest the authors focus on one or two major contributions instead of doing heavy parameter tuning via several different training tricks.
> > > 2. About the experimental numbers. Thanks for the clarification. Now they are clear. However, I suggest the authors stick to the original presentation of results to avoid confusion.
> > > 3. The visualizations are still not provided. The authors refer to the curves as visualizations. However, I wish to see visual comparisons between the predictions and ground truths instead of curves plotting the accuracies. If there are no visualizations showing the results, it is hard to interpret the numbers to understand the importance of the contributions.
> > > 4. Although I now understand the concept of "factorized," thanks to the author's response. However, the draft is still confusing about this core contribution of the paper:
> > > (1) The leading sentences of equation 8 say that "we apply the residual connections after ... input". In fact, the residual connection is irrelevant to the factorization at all. And if the authors wish to highlight their contribution to "factorization," they should use a separate paragraph saying what factorization is, how to understand the difference between factorization over the prior works, and what the motivations of factorizations are. All these important information hides behind vague sentences in different places. Therefore the readers can hardly get the points.
> > > (2) The sentence "the seemingly small change from R^l to R^l_d in the Fourier operator reduces the number of parameters to $O(LH^2MD)$" is also confusing. It is indeed a small change, actually, and the authors do not need to mention this fact here. It is, in fact, unclear to me why the original complexity is $M^D$. It seems unnecessary to incorporate some exponential complexity in a deep architecture. Actually, one possible explanation is $D$, and $M$ are both very small, so the baseline does not care about this detail. If so, the contribution looks even more incremental. Otherwise, the motivations are not motivated well. Unless there is a strong motivation behind this tiny change, it is not recommended to publish this on ICLR.
> > > (3) If the core contribution is factorization, equation 1-3  is not needed, and the model is not dependent on the PDE problem. Section 4 and the major background in Section 3 are still padding of the paper.
> > > 5. Figure 1 is much better than the previous version. Thanks.
> > > We will discuss this paper later, and I do not have more questions at this moment.
> > > jqw2

---

> > > > ### Author Response · Authors · 2022-11-23
> > > > **Clarification on Visualization and Factorization**
> > > >
> > > > Thank you for the further comments and questions.
> > > >
> > > > > The visualizations are still not provided
> > > >
> > > > Please refer to Figure A.3 (on page 14 in the Appendix, *not* in the main paper), which shows a heatmap of the vorticity field on the torus - here we show the ground truths, simulations from DNS, and simulations from F-FNO. The reviewer might have looked at the wrong figure.
> > > >
> > > > > It is, in fact, unclear to me why the original complexity is $M^D$
> > > >
> > > > As a concrete example, suppose we'd like to run a simulation on a 3D cube with 64 grid points on each spatial dimension. We also only take the top 16 modes, and there are 32 hidden dimensions. Here $M = 16, D=3, X=Y=Z=64, H=32$. In the baseline FNO:
> > > >
> > > > 1. The input has shape $(X, Y, Z, H) = (64, 64, 64, 32)$.
> > > > 2. We first take the Fourier transform, which also has the same shape $(64, 64, 64, 32)$.
> > > > 3. Next we keep only the top 16 Fourier modes, giving us an $(16, 16, 16, 32)$ array.
> > > > 4. In baseline FNO, we'd now multiply this array by a weight matrix of the shape $(16, 16, 16, 32, 32)$.
> > > >
> > > > Note how this weight matrix has $M^D \cdot H^2 = 16^3 \cdot 32^2 = 4,194,304$ parameters.
> > > >
> > > > In our F-FNO, we would instead perform the Fourier transform on each dimension separately. Thus instead of having one big matrix of shape $(16, 16, 16, 32, 32)$, we would now have three separate matrices of shape $(16, 32, 32)$. Thus in each F-FNO layer, we now only have $D \cdot M \cdot H^2 = 3 \cdot 16 \cdot 32^2 = 49,152$. Note that this is **a two-order of magnitude** reduction in model complexity.
> > > >
> > > > For actual numbers, please refer to, for instance, the 3D Plasticity results in Table A.6 (page 17 in the Appendix). In 3D Plasticity, we reduce the model complexity by roughly one order of magnitude, while also reducing the error by 60%.

---

> > > > > ### Comment · Reviewer_jqw2 · 2022-11-23
> > > > > **No figure A.4 can be found.**
> > > > >
> > > > > The authors say "See Figure A.4 in the revised submission"
> > > > > However, I find no Figure A.4 in the revised submission.
> > > > > I'm not sure whether this is a typo or the authors did not submit the revised version to the OpenReview system.
> > > > > Can the authors check this?
> > > > > Yours,
> > > > > jqw2

---

> > > > > > ### Author Response · Authors · 2022-11-23
> > > > > > **Re: No figure A.4 can be found**
> > > > > >
> > > > > > Apologies for the confusion. That was actually a typo in our response. We meant Figure A.3 in both the original and revised paper.
> > > > > >
> > > > > > (There was an earlier version where we added an extra figure in the Appendix, which pushed the original Figure A.3 to A.4. Afterwards, we decided to merge the extra figure into the new Figure 1 which visualizes the input/output of each dataset - but we forgot to update the reference in our response!)

---

> > > > > > > ### Comment · Reviewer_jqw2 · 2022-11-23
> > > > > > > **Thanks for clarification. Still confused about the visualization.**
> > > > > > >
> > > > > > > That makes sense. The authors claimed, "The reviewer might have looked at the wrong figure.", actually I did not find the A.4. and I did not know where to look.
> > > > > > > Let's see A.3, and I still think A.3 is not convincing.
> > > > > > > 1. Important baselines are missing from A.3. The key criticism about this draft is that it is incremental over FNO. However, after looking at A.3., I did not find FNO. In A.3., the authors only compare to DNS, actually, DNS is not a published paper and it is the naive approach of numerical simulation (just like the gt, it can only be a reference). So it is hard for the reviewer to decide whether it is superior to the state-of-the-art.
> > > > > > > 2. Why F-FNO 128X128 is correlated to DNS 2048X2048, not to DNS 128X128?
> > > > > > > 3. A.3 is not self-contained. For example, what is the unit and the meaning of the x, y axis? What is the unit of the red and green bar? These information should be given to make the figure rigorous.
> > > > > > > 4. The order of A.3 is wield: DNS (high RES), F-FNO, DNS (low RES). This is not easy to read.

---

> > > > > > > > ### Author Response · Authors · 2022-12-11
> > > > > > > > **Clarification on the visualization**
> > > > > > > >
> > > > > > > > Thank you for the feedback on Figure A.3 in the Appendix.
> > > > > > > >
> > > > > > > > > A.3 is not self-contained
> > > > > > > >
> > > > > > > > The heatmaps in this figure represent the surface of a torus mapped onto a 2D grid. The color in the heat maps represents the vorticity (the spinning motion) of the fluid on the torus. We will add these details to the figure caption.
> > > > > > > >
> > > > > > > > > Why F-FNO 128X128 is correlated to DNS 2048X2048, not to DNS 128X128?
> > > > > > > >
> > > > > > > > DNS 2048x2048 is treated as the ground-truth data. When running a simulation, the bigger the resolution is, the more accurate the simulation will be. 2048x2048 is the biggest resolution that can fit inside our 12GB GPU.
> > > > > > > >
> > > > > > > > Given that DNS 2048x2048 is the ground truth, the goal now is to find a solver that can produce results as close as possible to DNS 2048x2048, but in less time. In Figure A.3, we present two choices - either DNS 128x128 or F-FNO 128x128 (both of which are much faster than DNS 2048x2048; see Figure 4a).
> > > > > > > >
> > > > > > > > Figure A.3 shows that the neural operator F-FNO produces better results (correlating more with the ground truths) than DNS 128x128. In order words, for the same desired accuracy, F-FNO requires a smaller grid input than a numerical solver. This observation is backed up in Figure 4 (page 7).
> > > > > > > >
> > > > > > > > Thus "F-FNO 128X128 is correlated to DNS 2048X2048, not to DNS 128X128" simply means that F-FNO produces more accurate simulations.
> > > > > > > >
> > > > > > > > > Important baselines are missing from A.3.
> > > > > > > >
> > > > > > > > We intended Figure A.3 to serve as a visualization of one specific test example comparing F-FNO and DNS. In the next revision, we will add an extra figure in the Appendix to visualize one test example in Figure 3, comparing F-FNO with all the variants (FNO, FNO-TF, FNO-M, FNO-R, FNO++). We do, however, believe that one should not read too much from a visualization of a specific test example (hence our decision to put Figure A.3 in the Appendix). Instead, the quantitative results on the entire test set, along with the error bars, shown in Figure 3 and Table A.3, should serve as a more reliable indicator of quality.

---

> > > ### Comment · Reviewer_jqw2 · 2022-12-10
> > > **It's a pity that the authors do not response to me.**
> > >
> > > After discussing with other reviewers via the Zoom meeting, some of my questions are addressed, and it would be better if the authors clarify the mentioned points in the next revision. Also, some senior reviewers with great expertise in this domain insist that this paper has a clear empirical advantage over the baselines. Therefore, I agree to accept this paper right now. I don't think a simple method should not be accepted, whereas the AC also thinks so. I think the authors should clarify their central contribution (eq.8) in this case and not hide their contribution behind a set of tricks. I think a revision would be beneficial to the community and also to the authors.

---

> > > > ### Author Response · Authors · 2022-12-11
> > > > **Thank you for the feedback**
> > > >
> > > > Apologies for the late reply. We have now responded to the feedback on the visualization in the Appendix.
> > > >
> > > > Thank you also for the feedback on the factorization contribution. We will clarify Equation 8, discuss the concept of factorization in more detail, and make this contribution more prominent, in the next revision.

---

### Official Review · Reviewer_Uuaf · 2022-10-23

**Confidence:** 2
**Correctness:** 4
**Technical Novelty And Significance:** 3
**Empirical Novelty And Significance:** 2
**Recommendation:** 8

**Clarity, Quality, Novelty And Reproducibility:**

The paper is clearly written and the code provided has been structured well with good comment and documentation.


**Strength And Weaknesses:**

This paper has several aspects of strength:

1. Compared to [Li2021], the F-FNO framework has several changes: 1) a residual connection after the feedforward connection; 2) a spatial factorization; and 3) during the training, instead of unrolling the model, the F-FNO use teacher forcing and make online update based on Markov assumption;

2. The performance improvement of F-FNO over FNO is significant. Under several synthesis dataset  and using deep architectures, the reduction on MSE and time until correlation are both significant.

3. The introduction of forcing function in the F-FNO allows the model to take additional contextual information that helps to solve the equation.

This paper has several aspects for improvement:

1. While claiming to improve over the state-of-the-art algorithm, the experiments in the main paper only compare with FNO while the comparison with other methods are left in appendix. This is a missed opportunities since it is important to provide a comprehensive review on the related methods in both the parameter size, the MSE and time until correlation, as in [Li2021].  It is suggested to put some of the comprehensive study in the early main page.

2. The spatial factorization is a significant simplification for the model while it may over introduce additional bias since it implicitly assumes that the solution also factorizes spatially. Does it make sense physically in the scenario when the Navier-Stokes equations are applied ? What is the tradeoff for this factorization ?

3. As a PDE solver in the spectral domain, it is expected to discuss some tradeoff on the choice of spatial and spectral resolution (i.e. the design of the wavelength number vs. the dimension of input). It is known from Sampling Theorem that this choice of sampling rate would affect the performance of the system when the complex geometries require more detailed information to be kept after the discretization. This practical issue seems not discussed but it is interesting to have some thoughts on that.



**Summary Of The Paper:**

In this paper, the author proposee Factorized Fourier Neural Operator (F-FNO), a learning-based approach for simulating partial differential equations (PDEs). This work is seen as an improvement of the work by [Li2021] in which the Fourier Neural Operator was introduced. In this work, the author aims at improving the stability of the algorithm under complex geometries and noisy data by introducing more regularizations in the neural operator. The proposed F-FNO method is able to reduce the size of parameter by an order of magnitude while improving the performance over FNO significantly.


[Li2021] Zongyi Li, Nikola B. Kovachki, K. Azizzadenesheli, Burigede Liu, K. Bhattacharya, Andrew Stuart, and Anima Anandkumar. Fourier neural operator for parametric partial differential equations. In International Conference on Learning Representations, 2021a.


**Summary Of The Review:**

This paper has proposed an important update over existing Fourier Neural Operator and has brought it closer to practical use. With additional factorization, the F-FNO framework reduces the parameter size while attaining a great improvement in stability and performance when the layer of the neural network goes deep. This justifies my decision to accept this paper.

---

> ### Author Response · Authors · 2022-11-19
> **Response to Reviewer Uuaf**
>
> Thank you for your review.
>
> > (1) It is suggested to put some of the comprehensive study in the early main page.
>
> Figure 3 contains results against FNO and Crank-Nicholson solver. Figure 4 contains results against the Carpenter-Kennedy solver, and information about the trade-off between time until correlation and inference time. Table 2 contains results against geo-FNO.
>
> Due to the page limit, we leave many of the big tables to the Appendix. We opted for plots in the main paper to allow the reader to visualize key results quickly, leaving the details to the Appendix. Specifically, Table A.3 contains the raw numbers for Figure 3, and Tables A.4-6 expand on Table 2.
>
> > (2) The spatial factorization
>
> In our experiments, the only setting where F-FNO performs worse is when we only have 4 layers on the Elasticity dataset (see Table 2). Thus for very small networks, factorization might not give the model enough expressive power. The F-FNO shines in deep networks (8-24 layers) and in higher-dimensional problems (e.g., 3D Plasticity).
>
> Also note that in each F-FNO layer, we have a feedforward layer in the real space which gives the model a way to learn the interaction between the dimensions, mitigating potential bias from the factorization.
>
> Also see response (2) to Reviewer KRSZ.
>
> > (3) Sampling Theorem
>
> Thank you for the pointer to Sampling Theorem. We will consider it in future work.

---

### Official Review · Reviewer_n5hB · 2022-10-25

**Confidence:** 4
**Clarity, Quality, Novelty And Reproducibility:** The paper is clear and easy to follow…
**Correctness:** 3
**Technical Novelty And Significance:** 2
**Empirical Novelty And Significance:** 2
**Recommendation:** 5

**Strength And Weaknesses:**

[Strenghts]

- The main idea introduced by the authors, of factorizing the input dimension and treating the fourier transform of each dimension separately seems to work well for the various 2D Navier Stokes equations considered.
- The method seems to get better with size of depth, and requires less parameters to reach similar performance as normal FNO.

[Weakness]

- The main idea (of treating each dimension independently) is only testing for ONE family of PDEs and that too in just two dimensions. It is hard to know if factorizing will *always* perform better. For example, one scenario where it might not is for PDEs that have cross terms involving derivatives w.r.t different dimensions.
- Most of the contribution made by the authors seem to be utilizing training techniques (like teacher forcing) that have are shown to give better performance in other domains and test them for FNOs. This does not seem to be a very novel.
- The authors test F-FNO on irregular geometries, is it established that normal FNOs definitely do not do well of these irregular geometries considered by the authors. That comparison seems to be lacking in the paper.

**Summary Of The Paper:**

The paper introduces Factorized-FNO, where they consider separable Fourier representation (by taking the fourier transform of each dimension separately and independently) the authors achieve more “stable” models, whose performance increases as the networks are made deeper (something that does not happen with baseline FNOs), for 2D navier stokes equations.

The authors also improve the performance of original FNO baseline by introducing techniques like teacher forcing, a markov property.

They also test F-FNO on irregular geometries.

**Summary Of The Review:**

The authors have tested F-FNO on a single family of PDEs (and only in 2D) and it is hard to judge if it will always improve performance. The paper also lacks comparison with baselines (like FNO) for their abilities to hand irregular geometries.

---

> ### Author Response · Authors · 2022-11-19
> **Response to Reviewer n5hB**
>
> > (1) The main idea (of treating each dimension independently) is only testing for ONE family of PDEs and that too in just two dimensions. It is hard to know if factorizing will always perform better. For example, one scenario where it might not is for PDEs that have cross terms involving derivatives w.r.t different dimensions.
>
> Thank you for the opportunity to clarify. F-FNO is applied to three different PDEs and on geometries of different dimensionality. This prompted key changes 3, 4, and 5 in our Summary Response.
>
> Specifically, we test the F-FNO on three PDEs:
> 1) The incompressible Navier-Stokes equations on the torus (Equation 1 in the revised paper)
> 2) The equation of a solid body for Elasticity and Plasticity (Equation 2)
> 3) The Euler equations for Airfoil  (Equation 3)
>
> Furthermore, we test the F-FNO on three different geometries:
> 1) Regular grids on the torus (Figure 1a in the revised paper)
> 2) Point clouds on Elasticity (Figure 1b)
> 3) Structured meshes on Airfoil (Figure 1c) and Plasticity (Figure 1d)
>
> In particular, the Plasticity dataset is a time-dependent problem, in which the input is a 1D boundary condition and the output is 3D – a 2D mesh over 20 time steps. Results on Plasticity (see Table 2 and Table A.6) might partially address your concern since we need to apply each Fourier layer over both 2D space and 1D time.
>
> PDEs with cross terms involving derivatives wrt different dimensions are left as future work. Furthermore, we have not yet encountered a major PDE benchmark with cross terms. We appreciate being pointed in the right direction for potential future work.
>
> > (2) Most of the contribution made by the authors seem to be utilizing training techniques (like teacher forcing) that have are shown to give better performance in other domains and test them for FNOs. This does not seem to be a very novel.
>
> Training techniques are only one of our contributions. We have now clarified our contributions in the introduction:
>
> 1) *We propose a new representation, the F-FNO, which consists of separable Fourier representation and improved residual connections, reducing the model complexity and allowing it to scale to deeper networks (Fig. 2 and Eqs. (7) and (8)).*
> 2) *We improve the learning process of neural operators by incorporating training techniques such as Markov assumption, Gaussian noise, and cosine learning rate decay (Fig. 3); and investigate how well the operator can handle different input representations (Fig. 5).*
>
>
> Although on its own, each individual technique is an established technique, it did take a significant amount of careful experimentation to tune the neural operator. Please see our updated Figure 3 which outlines in detail the contribution from each training technique and change in model architecture. Overall there are six strategies that each had a positive effect and collectively reduced the error by 80%.
>
> > (3) The authors test F-FNO on irregular geometries, is it established that normal FNOs definitely do not do well of these irregular geometries considered by the authors. That comparison seems to be lacking in the paper.
>
> We’d like to clarify a potential source of confusion on the baselines FNO and geo-FNO. When [Li2021FNO] proposed the FNO, it only worked with regular grids because the key component (the discrete Fourier transform) can only be applied to a regular grid. Later [Li2022GeoFNO] adds an extra coordinate map to the FNO and call the resulting model geo-FNO. This coordinate map deforms an irregular real space into a regular computational space, thus allowing us to learn on irregular geometries such as point clouds. The rest of the geo-FNO architecture is identical to that of the FNO.
>
> In our work, we propose a new neural operator layer F-FNO. The F-FNO can work without the coordinate map on regular grids or with the coordinate map on irregular geometries. In both cases, we still keep the same name F-FNO.
>
> To summarize, it’s not possible to run the original FNO on irregular geometries without coordinate transformation. With the coordinate transformation, we’d just get geo-FNO.
>
> **References**
>
> [Li2021FNO] Zongyi Li, Nikola B. Kovachki, K. Azizzadenesheli, Burigede Liu, K. Bhattacharya, Andrew Stuart, and Anima Anandkumar. Fourier neural operator for parametric partial differential equations. In International Conference on Learning Representations, 2021a. URL https://openreview.net/forum?id=c8P9NQVtmnO.
>
> [Li2022GeoFNO] Zong-Yi Li, Daniel Z. Huang, Burigede Liu, and Anima Anandkumar. Fourier neural operator with learned deformations for pdes on general geometries. ArXiv, abs/2207.05209, 2022.

---

> > ### Comment · Reviewer_n5hB · 2022-11-24
> > **Thank you for your response**
> >
> > I would like to thank the authors for the rebuttal. At the outset, it is hard to judge the precise changes made by the authors since they are not clearly delineated (i.e they are not either written in a different color or any other means) which makes it hard to adjudge what precisely has changed. From the looks of it the three contributions laid out in the introduction seem to be pretty much overhauled.
> >
> > I still think that the architectural details and the other training techniques (known to perform better in general like teacher forcing, cosine lr and residual connections) is not very novel. (Residual connections helping for depth is not very surprising given vanishing gradients).
> >
> > I thank the authors for clarifying the points on complex geometries and how FNO can’t be a proper baseline, I presume geo-FNO is a more reasonable baseline which the authors have indeed compared against.
> >
> > I have increased this score to a 5. However, I still maintain that the contributions are not novel enough.

---

> > > ### Author Response · Authors · 2022-11-24
> > > **Re: Viewing Revision Diff**
> > >
> > > We thank the reviewer for the updated comments and score. Regarding the diff, at the top of this page, if we click on "Show Revision", then "Compare Revision", OpenReview will automatically generate a PDF diff for us at the following link:
> > >
> > > https://api.draftable.com/v1/comparisons/viewer/yVmSPr/uJVQXIfoHrRU

---

### Official Review · Reviewer_WWi7 · 2022-10-25

**Confidence:** 4
**Correctness:** 3
**Technical Novelty And Significance:** 3
**Empirical Novelty And Significance:** 3
**Recommendation:** 6

**Clarity, Quality, Novelty And Reproducibility:**

It is a well-written article with high quality. It seems a continuation of the authors' previous work but it is still very important.
I do not check the reproducibility given the constraint of time.

**Strength And Weaknesses:**

Pros:  The proposed method outperforms the state of art on PDEs with geometies and domains.
Cons: The literature reviews on the classical and recent PDE solvers are not very complete. I would like to see some comparison with the classical numerical method, such as the vortex model in approximating 2D Navier-Stokes/Euler with FFO.



**Summary Of The Paper:**

This paper uses the Fourier transform to learn neural operators that can handle long-range spatial dependencies. By factorizing the transform, using better residual connections, and improving the training setup, the proposed F-FNO outperforms the state of the art on PDEs on a variety of geometries and domains.

**Summary Of The Review:**

I would suggest to accept the current paper.

---

> ### Author Response · Authors · 2022-11-19
> **Response to Reviewer WWi7**
>
> Thank you for your review.
>
> > The literature reviews on the classical and recent PDE solvers are not very complete. I would like to see some comparison with the classical numerical method, such as the vortex model in approximating 2D Navier-Stokes/Euler with FFO.
>
> In our revised Section 2 (Related Work), we have now mentioned classical methods such as finite element methods, finite difference methods, finite volume methods, pseudo-spectral methods, Reynolds averaged Navier-Stokes, and large eddy simulation. Due to the page limit, we are not able to discuss these classical methods in detail. Our work focuses on building general neural operators which are not specific to any PDE. Thank you for the suggestion on the vortex model – we will look into it for future work.
>
> In Section 4 (Results), we have compared our model against two classical pseudo-spectral methods – Crank-Nicholson (for TorusLi, TorusVis, and TorusVisForce) and Carpenter-Kennedy (for TorusKochKov). These methods are in fact used to generate ground-truth data.

---

### Official Review · Reviewer_KRSZ · 2022-10-25

**Confidence:** 5
**Correctness:** 4
**Technical Novelty And Significance:** 3
**Empirical Novelty And Significance:** 4
**Recommendation:** 8

**Clarity, Quality, Novelty And Reproducibility:**

The paper is clearly written. It has good quality. The technical novelty is not very strong.

**Strength And Weaknesses:**

Strength:
- The work shows significant improvement on previous methods.
- The paper has a comprehensive study on many pde problems.
- It has a careful cost-accuracy study with the numerical solver. I especially love figure 4.

weak:
- The author improve FNO++ with many tricks. I assume F-FNO is also equipped with these tricks. It could be better to clearly list what these tricks are, and to what aspect they contribute to the overall improvement.
- It would be better to provide some intuition why the factorized structure help F-FNO scale with more layers.

**Summary Of The Paper:**

In this work, the authors proposed a novel neural operator architecture that factorizes the convolution on Fourier space into separate dimensions. Consequentially, the F-FNO model can scale up to a higher number of layers and achieve smaller errors. The paper has a comprehensive numerical study on multiple types of partial differential equations, considering chaotic systems and complex geometries. It also has a careful comparison with numerical solvers on the trade off between speed and accuracy.

**Summary Of The Review:**

Overall I find this paper interesting. It has a concrete contribution to the community. I recommend acceptance.

---

> ### Comment · Reviewer_KRSZ · 2022-11-04
> **Table 1**
>
> The geometric PDE datasets are probably taken from the Geo-FNO paper. In this case,
> - Airfoil: the input is the mesh grid (correct); the output should be the velocity fields $v$.
> - Plasticity: the input is the die function (correct, but better to write as the boundary condition); the output should be the displacement $u$.

---

> > ### Comment · Reviewer_KRSZ · 2022-12-10
> > **Table 1**
> >
> > Please remember to update table 1. The output of Airfoil should be the velocity fields v. The output of Plasticity should be the displacement u.

---

> > > ### Author Response · Authors · 2022-12-11
> > > **Re: Table 1**
> > >
> > > Thank you for the reminder. We will update Table 1 in the next revision.

---

> ### Author Response · Authors · 2022-11-19
> **Response to Reviewer KRSZ**
>
> Thank you for your review.
>
> > (1) The author improve FNO++ with many tricks. I assume F-FNO is also equipped with these tricks. It could be better to clearly list what these tricks are, and to what aspect they contribute to the overall improvement.
>
> Yes the F-FNO is equipped with these tricks. The tricks are mentioned on page 7 and consist of: *“normalizing the inputs, adding Gaussian noise to the training inputs, and using cosine learning rate decay”*. In the revised paper, we have summarized the training strategies in the abstract; and in both the contribution list and a new paragraph in the introduction.
>
> The F-FNO incorporates all features from our other variants (teacher forcing from FNO-TF, Markov assumption from FNO-M, residuals from FNO-R, and bag of tricks from FNO++). We have updated Figure 3 to make it clear how each change contributes to the improvement.
>
> > (2) It would be better to provide some intuition why the factorized structure helps F-FNO scale with more layers.
>
> [Kovachki2021] showed that neural operators are universal approximators for a sufficiently deep network. However, [Li2021FNO] and [Li2022GeoFNO] were only able to train a stable network at 4 layers. Due to the interaction terms, FNO and geo-FNO scale exponentially with the number of dimensions. We hypothesize that this explosion in parameters is one reason the neural operator becomes difficult to train at deep layers.
>
> In Section 3 and Equation 8, we show how the factorized structure allows us to remove an exponent from the model complexity, vastly reducing the number of parameters (e.g., see Table A.6 on how we reduce the parameters by an order of magnitude in 3D Plasticity).
>
> Note also that factorization by itself is not sufficient for scalability. From Figure 3, we can see that having better residual connections and incorporating various training techniques also contribute to the performance. Also see response (2) to reviewer Uuaf.
>
> **References**
>
> [Li2021FNO] Zongyi Li, Nikola B. Kovachki, K. Azizzadenesheli, Burigede Liu, K. Bhattacharya, Andrew Stuart, and Anima Anandkumar. Fourier neural operator for parametric partial differential equations. In International Conference on Learning Representations, 2021a. URL https://openreview.net/forum?id=c8P9NQVtmnO.
>
> [Li2022GeoFNO] Zong-Yi Li, Daniel Z. Huang, Burigede Liu, and Anima Anandkumar. Fourier neural operator with learned deformations for pdes on general geometries. ArXiv, abs/2207.05209, 2022.
>
> [Kovachki2021] Nikola Kovachki, Samuel Lanthaler, and Siddhartha Mishra. On universal approximation and error bounds for fourier neural operators. Journal of Machine Learning Research, 22(290):1–76, 2021a. URL http://jmlr.org/papers/v22/21-0806.html.

---

> ### Comment · Reviewer_KRSZ · 2022-12-10
> **Further comments**
>
> After discussing with other reviewers, I think it's important to emphasize the tricks should not be considered essential contributions of this work. Similar tricks like channel mixing and residual connection have been applied in previous works (https://arxiv.org/pdf/2111.13587.pdf). Indeed one can get error~5 on TorusLi dataset by simply adding LayerNorm.
>
> Still, the factorization improves the empirical performance with more layers, which seems to be significant.

---

### Comment · Reviewer_jqw2 · 2022-11-16
**No rebuttals?**

It seems that the authors do not post the rebuttals.

---

> ### Comment · Area_Chair_xb1d · 2022-11-16
> **Re: no rebuttals?**
>
> Thanks for the message!
>
> There is still time: they can post a response all the way through the discussion deadline (11/18).

---

> > ### Comment · Reviewer_jqw2 · 2022-11-16
> > **Thanks for your remind.**
> >
> > Got it!
> > I just received an email from the PCs saying, "discussion period is reaching the end ... please at the very least acknowledge that you have read their rebuttals." So I thought we should have read the rebuttals before Nov 18. It seems that after Nov 18, we cannot discuss this with the authors, if I understand that correctly and the authors do not wish to waive the discussion chance.

---

> > > ### Comment · Area_Chair_xb1d · 2022-11-16
> > > **Re: Thanks for your remind.**
> > >
> > > Correct, the main discussion period is until Nov 18. Subsequently the authors can respond to clarificatory questions from the reviewers if there are any, but will not be able to edit the submitted drafts.

---

> > > > ### Comment · Reviewer_jqw2 · 2022-11-16
> > > > **Thank you a lot!**
> > > >
> > > > Thank you again for the clarification of the policy!

---

> ### Author Response · Authors · 2022-11-16
> **Rebuttal is being prepared**
>
> Hi jqw2, we're still drafting up the rebuttal and revised paper. We will be posting them here before the deadline on 18 Nov.

---

### Author Response · Authors · 2022-11-19
**Summary Response**

We thank all reviewers for their questions and constructive feedback. Key changes in the revised submission include:

1) In Section 1, we have rewritten the introduction to clarify both our technical contribution and engineering contribution. Specifically:
    1) *We propose a new representation, the F-FNO, which consists of separable Fourier representation and improved residual connections, reducing the model complexity and allowing it to scale to deeper networks (Fig. 2 and Eqs. (7) and (8)).*
    2) *We improve the learning process of neural operators by incorporating training techniques such as Markov assumption, Gaussian noise, and cosine learning rate decay (Fig. 3); and investigate how well the operator can handle different input representations (Fig. 5).*

2) In Section 2, we have added a short comparison between classical methods and machine learning methods.

3) In Section 3, we did a major rewrite to clarify the range of PDEs being solved, restructured the model description so that our contributions in representation and training strategies are clearly visible, and clarified the difference between the two baselines (FNO and geo-FNO). These changes are also supported by major changes to Figure 1 and Figure 2 below.

4) Figure 1 has been revised to show the input and output of each of the four (PDE, geometry) combinations.

5) Figure 2 has been revised to show how F-FNO handles irregular geometries.

6) Error bars have been added to applicable figures.

7) Figure captions now describe evaluation settings.

8) We now clearly state that the N-MSE is shown as percentage in all figures, tables, and discussion text.

In addition we’d like to summarize a few key clarifications in relation to review comments and the changes above:

* Response (1) to reviewer n5hB clarifies that F-FNO is applied and evaluated on four different geometries with three different underlying PDEs. This prompted key changes 3, 4 and 5 above.

* Response (2) to reviewer n5hB and response (1.1) (1.3) (1.5) (1.6) to reviewer jqw2 clarify which contributions are made to neural operator representation and training strategies, respectively. This prompted key changes 1 and 3.

* Response (2.1)(2.3)(2.7) to reviewer jqw2 clarifies that our error metric is in percentages while the original baselines were reported as decimal (2.0 in our paper = 0.02 in [Li2021FNO]). This change is necessitated by the large performance improvements we saw, and trying to avoid too many leading zeros. We made key change 8 to clarify this in the revised paper.

---

### Decision · Program_Chairs · 2023-01-20

**Decision:**

Accept: poster

**Justification For Why Not Higher Score:**

The factorized architecture is a very simple change which yields non-trivial improvements. It deserves to be accepted at the conference, as FNOs are one of the dominant architectures for PDE solvers. On the other hand, the evaluation is fairly small-scale, so it's hard to say that the observations are very robust across PDE families.

**Justification For Why Not Lower Score:**

The main points of contention with the reviewers revolved around writing and clarity, which I believe can be easily addressed by the authors for the camera-ready version. The reviewers also added specific points to address in their edited reviews.

**Metareview: Summary, Strengths And Weaknesses:**

The paper proposes a relatively simple change to the recently proposed FNO architecture for PDE solvers. The main idea is to, rather than performing a FFT in each FNO layer, to perform an FFT on each of the coordinates separately, then apply the FNO-style trained kernel. While this change is relatively minor, it helps substantially with scaling the depth of the trained architectures, which in terms improves the trained model performance on various families of PDEs in consideration in the paper. The reviewers noted (both in Openreview, and during a Zoom meeting to discuss the paper), that the authors in the camera-ready version need to be forthright that the main original improvement is the factorization (eq. 8), whereas the current writing makes it seem like various other tricks known in the literature before (e.g. enforcing Markovianity, teacher forcing) are also part of the contributions. There were also several suggestions to clarify some plots in the figures (see the individual reviewer edits.)

**Note From Pc:**

if the above contains the word "oral" or "spotlight" please see: "oral" presentation means -> notable-top-5% and "spotlight" means -> notable-top-25%. As stated in our emails, we are disassociating presentation type from AC recommendations

**Summary Of Ac-Reviewer Meeting:**

The consensus was that while the main architectural innovation is very simple (the factorization in eq. 8), it has non-negligible benefits in scaling the depth of the architectures trained (which is a known issue for neural networks for PDE solvers), and yields notable benefits for the PDE families considered. The main sticking point were some writing issues:  the current writing makes it seem like various other tricks known in the literature before (e.g. enforcing Markovianity, teacher forcing) are also part of the contributions --- which they are not. I noted these points in the meta review, and the reviewers updated specific points in their reviews too.